# 3D-LLM: Injecting the 3D World into Large Language Models

**Yining Hong**
University of California, Los Angeles

**Haoyu Zhen**
Shanghai Jiao Tong University

**Peihao Chen**
South China University of Technology

**Shuhong Zheng**
University of Illinois Urbana-Champaign

**Yilun Du**
Massachusetts Institute of Technology

**Zhenfang Chen**
MIT-IBM Watson AI Lab

**Chuang Gan**
UMass Amherst and MIT-IBM Watson AI Lab

## Abstract

Large language models (LLMs) and Vision-Language Models (VLMs) have been proven to excel at multiple tasks, such as commonsense reasoning. Powerful as these models can be, they are not grounded in the 3D physical world, which involves richer concepts such as spatial relationships, affordances, physics, layout, and so on. In this work, we propose to inject the 3D world into large language models and introduce a whole new family of 3D-LLMs. Specifically, 3D-LLMs can take 3D point clouds and their features as input and perform a diverse set of 3D-related tasks, including captioning, dense captioning, 3D question answering, task decomposition, 3D grounding, 3D-assisted dialog, navigation, and so on. Using three types of prompting mechanisms that we design, we are able to collect over 1M 3D-language data covering these tasks. To efficiently train 3D-LLMs, we first utilize a 3D feature extractor that obtains 3D features from rendered multi-view images. Then, we use 2D VLMs as our backbones to train our 3D-LLMs. By introducing a 3D localization mechanism, 3D-LLMs can better capture 3D spatial information. Experiments on held-out evaluation dataset, ScanQA, SQA3D and 3DMV-VQA, outperform state-of-the-art baselines. In particular, experiments on ScanQA show that our model outperforms state-of-the-art baselines by a large margin (*e.g.*, the BLEU-1 score surpasses state-of-the-art score by 9%). Furthermore, experiments on our held-in datasets for 3D captioning, task composition, and 3D-assisted dialogue show that our model outperforms 2D VLMs. Qualitative examples also show that our model could perform more tasks beyond the scope of existing LLMs and VLMs. Project Page: : https://vis-www.cs.umass.edu/3dllm/.

## 1   Introduction

In the past several years, we have witnessed a surge of large language models (LLMs) (*e.g.*, GPT4 [33]) that excel at multiple tasks, such as communication and commonsense reasoning. Recent works have explored aligning images and videos with LLM for a new generation of multi-modal LLMs (*e.g.*, Flamingo [15], BLIP-2 [29]) that equip LLMs with the ability to understand and reason about 2D images. However, as powerful as the models can be in communication and reasoning,

37th Conference on Neural Information Processing Systems (NeurIPS 2023).

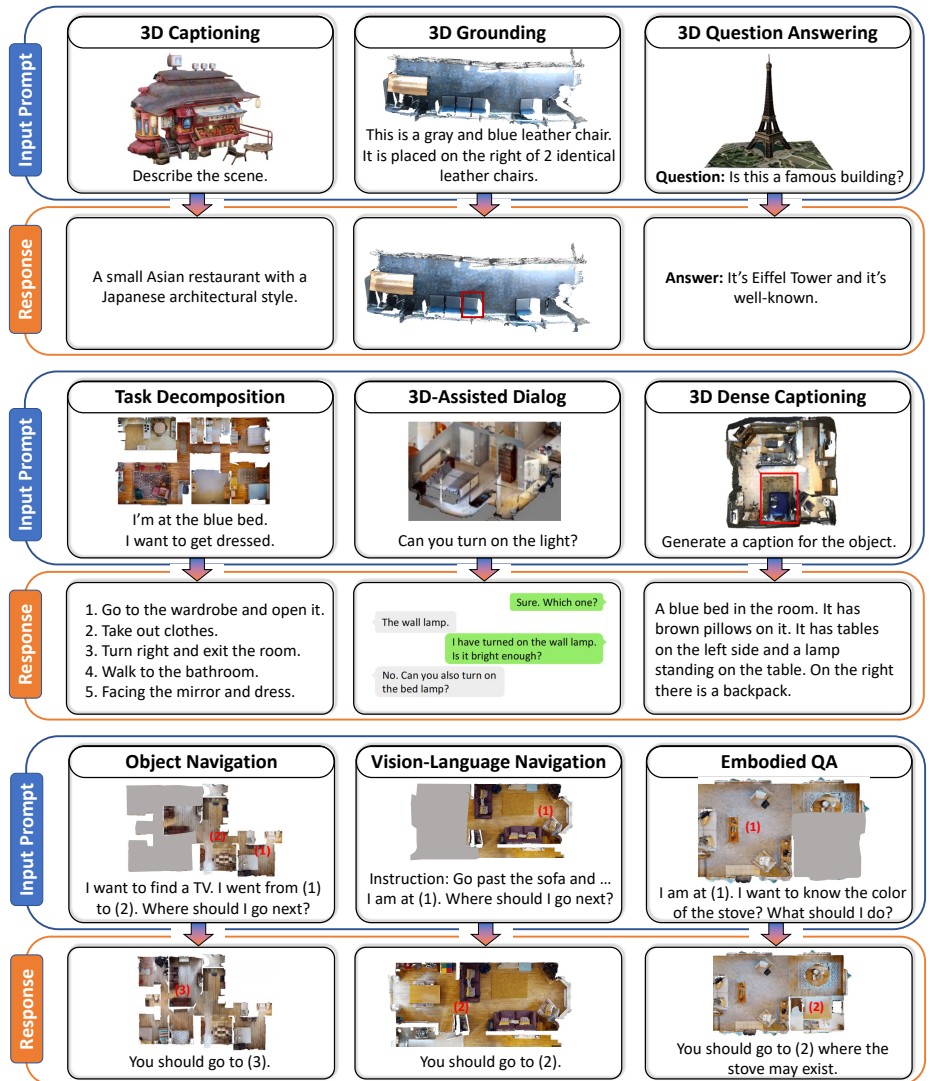

Figure 1: **Examples from our generated 3D-language data, which covers multiple 3D-related tasks.**

they are not *grounded in the real 3D physical world*, which involves richer concepts such as spatial relationships, affordances, physics and interaction so on. Therefore, such LLMs pale in comparison with the robots depicted in sci-fi movies - the assistants that could understand the 3D environments, as well as perform reasoning and planning based on the 3D understandings.

To this end, we propose to inject the 3D world into large language models, and introduce a whole new family of 3D-LLMs that could take 3D representations (*i.e.*, 3D point clouds with their features) as input, and perform a series of 3D-related tasks. By taking the 3D representations of scenes as input, LLMs are blessed with twofold advantages: (**1**) long-term memories about the entire scene can be stored in the holistic 3D representations, instead of episodic partial-view observations. (**2**) 3D properties such as affordances and spatial relationships can be reasoned from 3D representations, far beyond the scope of language-based or 2D image-based LLMs.

One major challenge of training the proposed 3D-LLMs lies in data acquisition. Unlike the vast amount of paired 2D-images-and-text data on the Internet, the scarcity of 3D data hinders the development of 3D-based foundation models. 3D data paired with language descriptions are even harder to obtain. To address this, we propose a set of unique data generation pipelines that could generate large-scale 3D data paired with language. Specifically, we make use of ChatGPT [33] and devise three efficient prompting procedures for communication between 3D data and language. In this way, we are able to obtain approximately one million 3D-language data covering a diverse set of

tasks, including but not limited to 3D captioning, dense captioning, 3D question answering, 3D task decomposition, 3D grounding, 3D-assisted dialog, navigation and so on, as shown in Figure 1.

The next challenge resides in how to obtain meaningful 3D features that could align with language features for 3D-LLMs. One way is to train 3D encoders from scratch using a similar contrastive-learning paradigm for the alignment between 2D images and language (*e.g.*, CLIP [36]). However, this paradigm consumes tremendous data, time, and GPU resources. From another perspective, there are numerous recent works that build 3D features from 2D multi-view images (*e.g.*, concept fusion [24], 3D-CLR [20]). Inspired by this, we also utilize a 3D feature extractor that constructs 3D features from the 2D pretrained features of rendered multi-view images. Recently, there are also quite a few visual-language models (*e.g.*, BLIP-2 [29], Flamingo [15]) utilizing the 2D pretrained CLIP features for training their VLMs. Since our extracted 3D features are mapped to the same feature space as 2D pretrained features, we can seamlessly use 2D VLMs as our backbones and input the 3D features for the efficient training of 3D-LLMs.

One crucial aspect of 3D-LLMs, different from vanilla LLMs and 2D VLMs, is that 3D-LLMs are expected to have an underlying 3D spatial sense of information. Thus, we develop a 3D localization mechanism that bridges the gap between language and spatial locations. Specifically, we append 3D position embeddings to the extracted 3D features to better encode spatial information. In addition, we append a series of location tokens to the 3D-LLMs, and localization can be trained via outputting location tokens given the language descriptions of specific objects in the scenes. In this way, 3D-LLMs could better capture 3D spatial information.

To sum up, our paper has the following contributions:

- We introduce a new family of 3D-based Large Language models (3D-LLMs) that can take 3D points with features and language prompts as input, and perform a variety of 3D-related tasks. We focus on tasks beyond the scope of vanilla LLMs or 2D-LLMs, such as tasks about holistic scene understanding, 3D spatial relationships, affordances and 3D planning.

- We devise novel data collection pipelines that could generate large-scale 3D-language data. Based on the pipelines, we collect a dataset that has over 1M 3D-language data that cover a diverse set of 3D-related tasks, including but not limited to 3D captioning, dense captioning, 3D question answering, task decomposition, 3D grounding, 3D-assisted dialog, navigation, and so on.

- We use a 3D feature extractor that extracts meaningful 3D features from rendered multi-view images. We utilize 2D pretrained VLMs as our backbones for efficient training. We introduce a 3D localization mechanism for training the 3D-LLMs to better capture 3D spatial information.

- Experiments on held-out evaluation dataset, ScanQA, SQA3D and 3DMV-VQA, outperform state-of-the-art baselines. In particular, 3D LLMs outperform baselines by a large margin on ScanQA (*e.g.,* 9% for BLEU-1 and 10% for CIDER). Experiments on held-in datasets for 3D captioning, task composition, and 3D-assisted dialogue show that our model outperforms 2D VLMs. Qualitative studies further demonstrate that our model is able to handle a diverse set of tasks.

- We release our 3D-LLMs, the 3D-language dataset, and language-aligned 3D features of the dataset for future research development [1].

## 2 Related Works

**Large Language Models.** Our work is closely related to large language models [4, 14, 37, 10, 34] (LLMs) like GPT-3 [4] and PaLM [10], which are able to handle different language tasks with a single model and show strong generalization abilities. These models are typically trained on massive textual data with self-supervised training targets like predicting the next tokens [4, 37] or reconstructing the masked tokens [14, 38]. To better align these LLMs' predictions to human instructions, improve the models' generalization abilities on unseen tasks, a series of instruction tuning methods [35, 42] and datasets [11, 13] have been proposed. In this work, we aim to inject the 3D world into large language models, understanding rich 3D concepts such as spatial relations, affordances, and physics.

**Vision-Language Pre-trained Models.** Our work is also related to vision-language pre-trained models that connect images and natural language [30, 31, 18, 36, 25]. Some research [36, 25] learn to train models from scratch with massive image-language pairs and apply them to downstream tasks like visual question answering [19, 47], captioning [7], and referring expression comprehension [46] with finetuning. Other researchers have connected pre-trained vision models and pre-trained LLMs

---

[1]https://github.com/UMass-Foundation-Model/3D-LLM

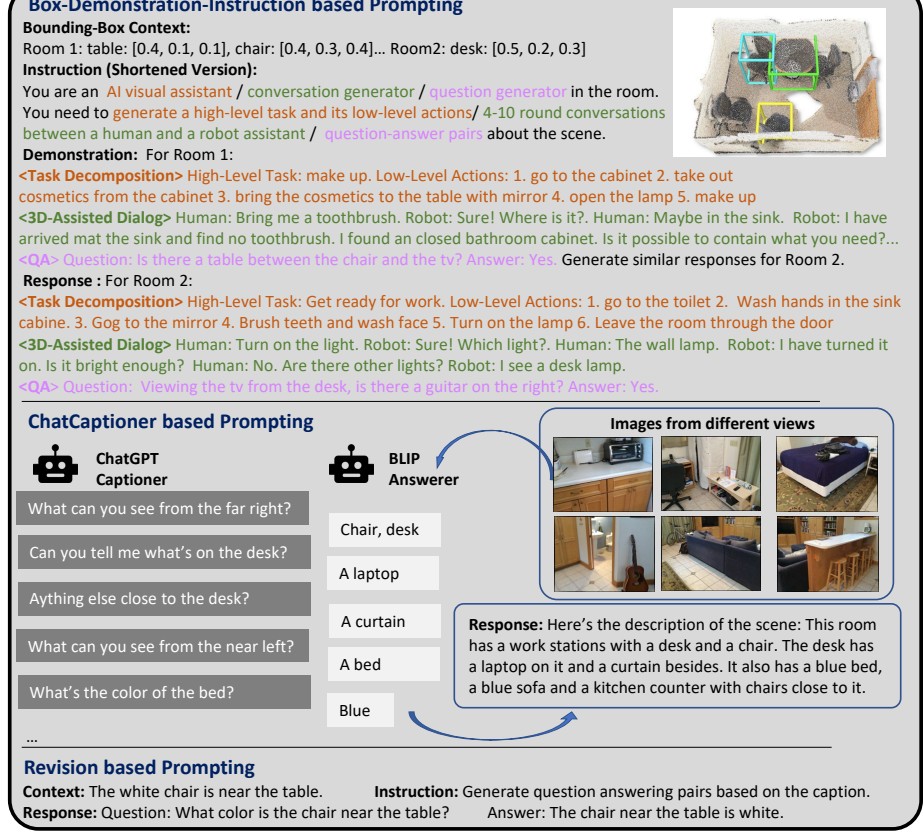

Figure 2: **3D-language data generation pipelines.**

with additional learnable neural modules like perceiver [2] and QFormers [30], leveraging perception abilities in pre-trained vision models, and reasoning and generalization capacities in LLMs. Inspired by these previous works, we plan to build an AI assistant that could understand the 3D world and perform corresponding 3D reasoning and planning. This is not trivial and we need to overcome obstacles like how to handle the problem of data sparsity, how to align the 3D world with 2D images, and how to capture 3D spatial information.

**3D & Language.** Another line of research that is similar to ours is 3D and language [5, 45, 8, 20, 1, 16, 22, 45, 3]. ScanQA [45] requires a model to answer questions related to the 3D world; ScanRefer [5] asks a model to localize a region that the text expression refer to; 3D captioning [8] tests models' abilities to generate captions describing the 3D scenes. However, these 3D tasks and their corresponding models are usually task-specific and could only handle cases within the same distribution of the training sets without generalization. Different from them, we aim to build a 3D model that could handle different tasks at the same time and enable new abilities like 3D-assistant dialog and task decomposition.

## 3 3D-Language Data Generation

The community has witnessed the proliferation of multi-modal data thanks to easy access to a tremendous amount of 2D image and text pairs on the internet. However, when it comes to 3D-related data, obtaining multimodal resource is not easy, due to not only the scarcity of 3D assets, but also the difficulty of providing language data for 3D assets. There are some existing datasets that contain 3D-language data (*e.g.*, ScanQA [45], ScanRefer [5]). However, they are limited with regard to both quantity and diversity, restricted to only one task per dataset. How to generate a 3D-language dataset that can be utilized for all kinds of 3D-related tasks is well worth delving into.

Inspired by the recent success of large language models like GPT [33], we propose to leverage such models for 3D-language data collection. Specifically, as shown in Figure 2, we have three ways to prompt a text-only GPT for generating data. 1) boxes-demonstration-instruction based prompting. We input the axis-aligned bounding boxes (AABB) of both the rooms and the objects in the 3D

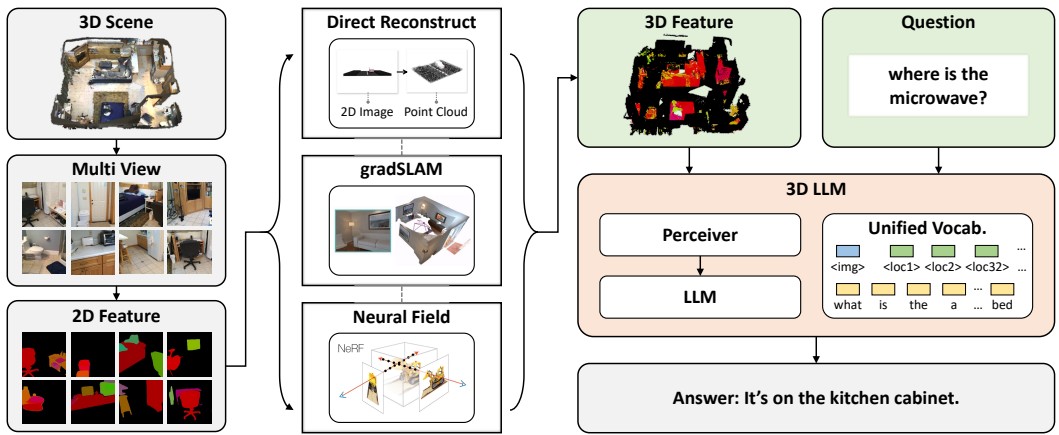

Figure 3: **Overview of our 3D-LLM framework.** The first two columns show our 3D feature extractor. We first render a few multi-view images from the 3D scene, extract 2D dense features, and then construct 3D features from these multi-view images using three kinds of methods. And then, the 3D features and input language prompts are input to the 3D-LLMs to generate responses.

scenes, providing information about the semantics and spatial locations of the scene. We then provide specific instructions to the GPT model to generate diverse data. We give 0-3 few-shot demonstration examples of the GPT model showing what kind of data it is instructed to generate. 2) ChatCaptioner based prompting. We utilize techniques similar to [48], in which ChatGPT is prompted to ask a series of informative questions about an image and BLIP-2 [29] answers the questions. In order to collect 3D-related data, we first sample several images from different views of a 3D scene. These images are fed into ChatGPT and BLIP-2 to get the caption of each image. We then leverage ChatGPT to summarize all these captions, which contain information about different regions, to form a global 3D description of the entire scene. 3) Revision based prompting. It can be used to transfer one type of 3D data to another.

Given the prompting pipelines, GPT is able to generate various types of 3D-language data as summarized in Figure 1. More data generation details and prompt designs are shown in the Appendix.

We mainly establish our 3D-language dataset upon several 3D assets:

- Objaverse is a universe of 800K 3D objects. However, since the language descriptions were extracted from online sources and not examined by humans, most objects have very noisy descriptions (*e.g.,* with urls) or no descriptions. We utilize ChatCaptioner based prompting to generate high-quality 3D-related descriptions for the scenes and reivison-based prompting to generate questions.
- Scannet [12] is a richly-annotated dataset of approximately 1k 3D indoor scenes. It provides semantics and bounding boxes of the objects in the scenes.
- Habitat-Matterport (HM3D) [39] is a dataset of 3D environments of embodied AI. HM3DSem [44] further adds semantic annotations and bounding boxes for more than 200 scenes of HM3D. We use the pre-segmented rooms of HM3D in 3D-CLR [20].

## 4 3D-LLM

### 4.1 Overview

In this section, we introduce how we train our 3D-LLMs. We argue that it's hard to train 3D-LLMs from scratch, since our collected 3D-language dataset is still not the size of billion-scale image-language dataset used to train 2D VLMs. Furthermore, for 3D scenes, there are no available pretrained encoders like those for 2D images (e.g., CLIP ViT encoders). Thus, retraining 3D-language models from scratch is data-inefficient and resource-heavy. Recently, researchers have proposed to extract 3D features from 2D multi-view images [24, 20]. Using these alignment methods, we could use pretrained image encoders to extract image features, and then map the features to the 3D data. Since the pretrained image features serve as inputs to 2D VLMs, the mapped 3d features of the same feature space can also be seamlessly fed into the pretrained 2D VLMs, which we use as our backbones to train 3D-LLMs. We also propose a 3D localization mechanism to boost the model's ability to capture 3D spatial information. Figure 3 shows our framework.

## 4.2 3D Feature Extractor

The first step of training 3D-LLMs is to build meaningful 3D features that could be aligned with language features. For 2D images, there exist feature extractors like CLIP, which learn visual models from language supervision. The models are pretrained using billion-scale internet data of image-language pairs. It's hard to pre-train such feature learners from scratch, since there are no 3D-language assets comparable to internet-scale image-language pairs in terms of quantity and diversity.

On the contrary, numerous methods have been proposed to extract 3D features from 2D multi-view images [24, 20, 17, 21]. Inspired by these works, we extract features for 3D points by rendering the 3D scenes in several different views, and construct 3D features from rendered image features.

We first extract pixel-aligned dense features for rendered images following [24]. Then, we utilize three methods to construct 3D features from rendered image features. These methods are designed for different types of 3D data.

- **Direct Reconstruction.** We directly reconstruct point cloud from rgbd images rendered from the 3D data using ground-truth camera matrixes. The features are directly mapped to the reconstructed 3D points. This method is suitable for rendered rgbd data with perfect camera poses and intrinsics.

- **Feature Fusion.** Similar to [24], we fuse 2D features into 3D maps using gradslam [27]. Different from dense mapping methods, the features are fused in addition to depths and colors. This method is suitable for 3D data with noisy depth map renderings, or noisy camera poses and intrinsics.

- **Neural Field.** We utilize [20], which constructs 3D compact representation using neural voxel field [40]. Specifically, each voxel in the field has a feature in addition to density and color. Then we align 3D features in the rays and 2D features in the pixels using MSE loss. This method is for 3D data with RGB renderings but no depth data, and noisy camera poses and intrinsics.

In this way, we are able to obtain the $< N, \mathcal{D}_v >$-dim 3D features of each 3D scene, where $N$ is the number of points in the point cloud, and $\mathcal{D}_v$ is the feature dimension.

## 4.3 Training 3D-LLMs

### 4.3.1 2D VLMs as backbones

In addition to the feature extractor, training 3D-LLMs from scratch is also non-trivial. In fact, according to [29, 15], the training of 2D VLMs only begins to show "signs of life" after consuming half a billion images. They usually use frozen and pre-trained image encoders such as CLIP to extract features for 2D images. Considering that with 3D feature extractor, the 3D features can be mapped into the same feature space as 2D images, it's reasonable to use these 2D VLMs as our backbones.

The perceiver architecture proposed by [23] leverages an asymmetric attention mechanism to iteratively distill inputs into a tight latent bottleneck, allowing it to handle very large inputs of arbitrary input sizes, thus can tackle different modalities. This architecture is utilized in VLMs like Flamingo [15]. BLIP-2 [29] also utilizes a similar structure called QFormer. The 2D image features, output from frozen image encoders, are flattened and sent to the perceiver to generate a fixed-sized input. Given that our 3D features are in the same feature space as the 2D features by the 3D feature extractor, and that perceiver is able to handle inputs of arbitrary input sizes of the same feature dimension, point cloud features with arbitrary sizes could also be fed into the perceiver. Therefore, we use the 3D feature extractor to extract the 3D features in the same feature space as the features of the frozen image encoders. Then, we use pretrained 2D VLMs as our backbones, input the aligned 3D features to train 3D-LLMs with our collected 3D-language dataset.

### 4.3.2 3D Localization Mechanism

Notice that since 3D features are reconstructed via 2D pretrained feature extractor that has been aligned with language (*e.g.,* CLIP [36] and EVA-CLIP [41]), localization can be performed by directly calculating the similarity between 3D features and language features. However, Apart from building 3D features, which can be aligned with language semantics, it's also essential that the model itself could capture 3D spatial information. To this end, we propose a 3D localization mechanism that boosts 3D LLMs' abilities to absorb spatial information. It consists of two parts:

**Augmenting 3D features with position embeddings** Besides the 3D features aggregated from 2D multi-view features, we also add position embeddings to the features. Supposing the feature dim is $\mathcal{D}_v$, we generate sin/cos position embeddings of the three dimensions, each has an embedding size

$\mathcal{D}_v/3$. We concatenate the embeddings of all three dimensions, and add them to the 3D features with a weight.

**Augmenting LLM vocabularies with location tokens** In order to align 3D spatial locations with LLMs, we propose to embed 3D locations in the vocabularies, following [6] and [43]. To be specific, the region to be grounded can be denoted as a sequence of discrete tokens representing the bounding box in the form of AABB. The continuous corner coordinates of the bounding boxes are uniformly discretized to voxel integers as location tokens $\langle x_{min}, y_{min}, z_{min}, x_{max}, y_{max}, z_{max} \rangle$. After adding these additional location tokens, we unfreeze the weights for these tokens in the input and output embeddings of language models.

## 5 Experiments

We first introduce the architecture, and training and evaluation protocols. In Sec 5.1, we analyze the held-out experiments on ScanQA [3], SQA3D [32], and 3DMV-VQA [20] Dataset. Sec 5.2 covers more analysis on held-in evaluation and qualitative examples.

**Architecture** We experiment on three backbone 2D VLMs for 3D-LLMs: Flamingo 9B, BLIP-2 Vit-g Opt2.7B, BLIP-2 Vit-g FlanT5-XL. For BLIP-2, during pre-training the 3D-LLMs, we initialize the model from BLIP-2 checkpoints released in LAVIS library [28], and finetune the parameters for the QFormer. 3D features are 1408-dim features, same as EVA_CLIP [41] hidden feature dim used by BLIP-2. We keep most parts of the LLMs (*i.e.,* Opt and FlanT5) frozen, except the weights for the newly-added location tokens in the input and the output embeddings. For Flamingo, we initialize the model from the Flamingo9B checkpoint released in OpenFlamingo repository [2]. We finetune the parameters for perceiver, gated cross attention layers, and the weights for additional location tokens in the input and output embeddings. 3D features are 1024-dim features, same as CLIP hidden feature dim used by Flamingo. For generating class-agnostic (generic) object masks for the 2D pixel-aligned dense feature extraction, we follow [24] and use the Mask2Former (M2F) [9] or the segment anything (SAM) [26].

**Training & Evaluation Datasets & Protocols** We split our datasets into two genres, held-in datasets and held-out datasets. Specifically, our 3D-language data generation pipeline generates the held-in datasets of multiple tasks. We utilize training sets of held-in datasets for pre-training foundation 3D-LLMs, and their validation sets can be applied for held-in evaluation. During pre-training, we mix the held-in datasets of all tasks. The models are trained with the standard language modeling loss to output responses. Held-out datasets, on the other hand, are not used in training the foundation 3D-LLMs. We use three held-out 3D question answering datasets for held-out evaluation: ScanQA, SQA3D and 3DMV-VQA.

### 5.1 Held-Out Evaluation

#### 5.1.1 Experiments on ScanQA

We finetune our pretrained 3D-LLMs on the ScanQA dataset and compare with baseline models.

**Baselines & Evaluation Metrics** We include representative baseline models on the benchmark. **ScanQA** is the state-of-the-art method on the benchmark that uses VoteNet to obtain object proposals, and then fuse them with language embeddings. **ScanRefer+MCAN** is a baseline that identifies the referred object and the MCAN model is applied to the image surrounding the localized object. **VoteNet+MCAN** detects objects in a 3D space, extracts their features, and uses them in a standard VQA model. Notably, these baseline models all extract explicit object representations from a pretrained localization module. In addition to these baselines, we also design several LLM-based baselines. **LLaVA** is a visual instruction tuning that connects a vision encoder and LLM for general-purpose visual and language understanding. We use its pretrained model and do zero-shot evaluation on our dataset. We use a single random image as input. We use LLaVA 13B model. **ULIP encoders + LLMs** use existing pre-trained 3D encoders with LLMs, for comparison between 3D pre-trained encoders, and 2D encoders for feature encoding. **Single Image + Pretrained VLMs** use our 2D VLM backbones (*i.e.*, flamingo and BLIP-2), replace the 3D inputs of 3D-LLMs with single image features to train the models, and then finetune on ScanQA dataset. **Multi-View Image + Pretrained VLMs** use our 2D VLM backbones, replace the 3D inputs of 3D-LLMs with concatenated features of multi-view images to train the models, and then finetune on ScanQA dataset. We report BLEU, ROUGE-L, METEOR, CIDEr for robust answer matching. We also use exact match (EM) metric.

|  | B-1 | B-2 | B-3 | B-4 | METEOR | ROUHE-L | CIDER | EM |
|---|---|---|---|---|---|---|---|---|
| VoteNet+MCAN* | 28.0 | 16.7 | 10.8 | 6.2 | 11.4 | 29.8 | 54.7 | 17.3 |
| ScanRefer+MCAN* | 26.9 | 16.6 | 11.6 | 7.9 | 11.5 | 30 | 55.4 | 18.6 |
| ScanQA* | 30.2 | 20.4 | 15.1 | 10.1 | 13.1 | 33.3 | 64.9 | 21.0 |
| LLaVA(zero-shot) | 7.1 | 2.6 | 0.9 | 0.3 | 10.5 | 12.3 | 5.7 | 0.0 |
| ULIPPointMLP+flant5 | 18.4 | 7.2 | 2.7 | 1.4 | 7.4 | 18.1 | 26.9 | 7.5 |
| ULIPPointMLP+opt | 19.1 | 7.3 | 2.7 | 1.9 | 7.4 | 18.2 | 28.0 | 8.4 |
| ULIPPointBERT+flant5 | 29.2 | 17.9 | 10.3 | 6.1 | 11.6 | 28.1 | 50.9 | 14.5 |
| ULIPPointBERT+opt | 28.8 | 16.9 | 9.7 | 5.9 | 11.3 | 27.9 | 50.5 | 13.8 |
| flamingo-SingleImage | 23.8 | 14.5 | 9.2 | 8.5 | 10.7 | 29.6 | 52 | 16.9 |
| flamingo-MultiView | 25.6 | 15.2 | 9.2 | 8.4 | 11.3 | 31.1 | 55 | 18.8 |
| BLIP2-flant5-SingleImage | 28.6 | 15.1 | 9.0 | 5.1 | 10.6 | 25.8 | 42.6 | 13.3 |
| BLIP2-flant5-MultiView | 29.7 | 16.2 | 9.8 | 5.9 | 11.3 | 26.6 | 45.7 | 13.6 |
| 3D-LLM (M2F, flamingo) | 30.3 | 17.8 | 12.0 | 7.2 | 12.2 | 32.3 | 59.2 | 20.4 |
| 3D-LLM (M2F, BLIP2-opt) | 35.9 | 22.5 | 16.0 | 9.4 | 13.8 | 34.0 | 63.8 | 19.3 |
| 3D-LLM (SAM, BLIP2-opt) | 35.0 | 21.7 | 15.5 | 9.5 | 14.0 | 34.5 | 67.1 | 19.8 |
| 3D-LLM (M2F, BLIP2-flant5) | **39.3** | **25.2** | **18.4** | 12.0 | 14.5 | 35.7 | 69.4 | 20.5 |
| 3D-LLM (SAM, BLIP2-flant5) | 37.5 | 24.1 | 17.6 | **12.9** | **15.1** | **37.5** | **74.5** | **21.2** |

Table 1: Experimental results on ScanQA validation set. * Means the models use explicit object representations. B-1, B-2, B-3, B-4 denote BLEU-1, BLEU-2, BLEU-3, BLEU-4 respectively. M2F denotes mask2former, SAM denotes Segment Anything.

|  | Format | test set | | | | | | Avg. |
|---|---|---|---|---|---|---|---|---|
|  |  | What | Is | How | Can | Which | Others |  |
| Blind test | SQ → A | 26.75 | 63.34 | 43.44 | 69.53 | 37.89 | 43.41 | 43.65 |
| ScanQA(w/o $s^{\text{txt}}$) | VQ → A | 28.58 | 65.03 | **47.31** | 66.27 | 43.87 | 42.88 | 45.27 |
| ScanQA | VSQ → A | 31.64 | 63.80 | 46.02 | 69.53 | 43.87 | 45.34 | 46.58 |
| ScanQA+aux task | VSQ → AL | 33.48 | **66.10** | 42.37 | **69.53** | 43.02 | 46.40 | 47.20 |
| MCAN | VSQ → A | 28.86 | 59.66 | 44.09 | 68.34 | 40.74 | 40.46 | 43.42 |
| ClipBERT | VSQ → A | 30.24 | 60.12 | 38.71 | 63.31 | 42.45 | 42.71 | 43.31 |
| Unified QA | VSQ → A | 33.01 | 50.43 | 31.91 | 56.51 | 45.17 | 41.11 | 41.00 |
| Unified QA | VSQ → A | 27.58 | 47.99 | 34.05 | 59.47 | 40.91 | 39.77 | 38.71 |
| GPT-3 | VSQ → A | 39.67 | 45.99 | 40.47 | 45.56 | 36.08 | 38.42 | 41.00 |
| GPT-3 | VSQ → A | 28.90 | 46.42 | 28.05 | 40.24 | 30.11 | 36.07 | 34.57 |
| 3D-LLM | VSQ → A | **37.05** | 65.18 | 45.81 | 67.46 | **51.00** | **49.82** | **49.79** |

Table 2: Experimental Results on SQA3D test set. In the Format column, "V" means the 3D visual inputs, "S" means the situation inputs, "Q" and "A" denote questions and answers respectively. Here we use 3D-LLM (SAM, BLIP2-flant5).

**Result Analysis** We report our results on ScanQA validation set in Table 1. We observe a significant increase in the evaluation metrics. For example, for BLEU-1, our model outperforms the state-of-the-art ScanQA model by ∼9% for validation set. For CIDER, we report a ∼10% gain compared to ScanQA, and much higher than other 3D-based baselines. These results show that by injecting 3D into LLMs, the models can generate answers that are much more similar to the ground-truth answers. Furthermore, 3D-based baselines use object detectors like VoteNet to segment the objects, and then send per-object features into their models, while our inputs are holistic 3D features without explicit object representations. This shows that our model could perform visual reasoning about objects and their relationships even without explicit object representations. We then examine whether 2D VLMs have the same ability. We find that by taking single-view images or multi-view images as inputs, the performances drop much compared to 3D-LLMs. Specifically, multi-view images also contain information about the whole scene. However, they have much lower performances compared to 3D-LLMs, probably because features of multi-view images are disorganized, thus losing 3D-related information.

### 5.1.2 Experiments on SQA3D

SQA3D [32] requires the tested agent to first understand its situation (position, orientation, etc.) in the 3D scene as described by text, then reason about its surrounding environment and answer a question under that situation. We finetune our pretrained 3D-LLMs on the SQA3D dataset and compare with baseline models. We include all baseline models introduced by the original paper. Specifically, ScanQA+aux task achieves the SOTA performance by adding two auxilliary tasks: prediction the

position and rotation of the agent situation. Table 2 shows the results. We can see that our 3D-LLM outperforms all baseline models a lot, even without training with auxiliary tasks and losses.

| Methods | Concept | Counting | Relation | Comparison | Overall |
|---|---|---|---|---|---|
| NS-VQA* | 59.8 | 21.5 | 33.4 | 61.6 | 38.0 |
| 3D-Feature+LSTM | 61.2 | 22.4 | 49.9 | 61.3 | 48.2 |
| 3D-CLR* | 66.1 | **41.3** | 57.6 | 72.3 | 57.7 |
| flamingo-SingleImage | 58.7 | 18.5 | 38.4 | 60.1 | 40.3 |
| flamingo-MultiView | 60.0 | 18.3 | 40.2 | 61.4 | 41.6 |
| BLIP-SingleImage | 58.0 | 20.4 | 42.3 | 62.3 | 43.1 |
| BLIP-MultiView | 61.9 | 21.1 | 48.0 | 62.3 | 47.1 |
| 3D-LLM (M2F, flamingo) | 68.9 | 32.4 | 61.6 | 68.3 | 58.6 |
| 3D-LLM (M2F, BLIP2-opt) | 63.4 | 30.7 | 57.6 | 65.2 | 54.9 |
| 3D-LLM (SAM, BLIP2-opt) | 73.4 | 24.5 | 63.2 | 77.6 | 61.5 |
| 3D-LLM (M2F, BLIP2-flanT5) | 68.1 | 31.4 | 55.1 | 69.7 | 54.6 |
| 3D-LLM (SAM, BLIP2-flanT5) | **76.3** | 30.2 | **64.3** | **80.2** | **64.0** |

Table 3: Experimental results on 3DMV-VQA dataset. * denotes using explicit object representations and neuro-symbolic reasoning.

### 5.1.3 Experiments on 3DMV-VQA

We finetune our pretrained 3D-LLMs on the 3DMV-VQA dataset and compare with baseline models. We include all baseline models introduced by the original paper. Specifically, 3D-CLR [20] is the SOTA achieves the SOTA performance via neuro-symbolic reasoning based on 3D features.

**Result Analysis** Table 3 shows the performances on 3DMV-VQA. We can see that 3D-LLMs outperform state-of-the-art baseline model in the question types of concept and relation, and also in the overall performance. Our model also outperforms 3D-Feature+LSTM, demonstrating the power of LLMs over vanilla language models with similar 3D features as inputs. Overall, 3D-based methods outshine 2D-based versions of the methods. Our 3D-LLMs outperform their corresponding 2D VLMs with image input, further demonstrating the importance of 3D representations for 3D-LLMs.

| Tasks | Models | BLEU-1 | BLEU-2 | BLEU-3 | BLEU-4 | METEOR | ROUGH-L |
|---|---|---|---|---|---|---|---|
| | flamingo-SingleImage | 29.0 | 17.9 | 12.5 | 12.1 | 12.4 | 28.2 |
| | flamingo-MultiView | 29.5 | 18.6 | 13.7 | 12.4 | 14.0 | 29.0 |
| | BLIP2-flant5-SingleImage | 30.3 | 18.3 | 14.5 | 12.0 | 13.1 | 30.9 |
| 3D Captioning | BLIP2-flant5-MultiView | 34.4 | 23.9 | 18.0 | 14.1 | 17.5 | 35.7 |
| | 3D-LLM (flamingo) | 36.1 | 24.5 | 18.7 | 15.6 | 17.6 | 35.8 |
| | 3D-LLM (BLIP2-opt) | 35.7 | 26.7 | 20.3 | 15.9 | 18.7 | 40.1 |
| | 3D-LLM (BLIP2-t5) | 39.8 | 31.0 | 24.7 | 20.1 | 17.7 | 42.6 |
| | 3D-LLM (SAM, BLIP2-t5) | **44.5** | **38.6** | **29.5** | **24.2** | **22.1** | **45.4** |
| | flant5 | 27.4 | 16.5 | 11.1 | 8.7 | 9.5 | 27.5 |
| | flamingo-SingleImage | 29.4 | 18.7 | 11.3 | 9.4 | 10.0 | 26.8 |
| | flamingo-MultiView | 30.6 | 21.3 | 11.9 | 9.1 | 10.4 | 27.9 |
| | BLIP2-flant5-SingleImage | 28.4 | 17.3 | 10.6 | 9.1 | 10.2 | 27.4 |
| 3D-assisted Dialog | BLIP2-flant5-MultiView | 32.4 | 20.9 | 12.1 | 9.5 | 11.0 | 29.5 |
| | 3D-LLM (flamingo) | 35.0 | 22.8 | 15.4 | 10.6 | 16.0 | 34.2 |
| | 3D-LLM (BLIP2-opt) | 39.6 | 27.5 | 20.5 | 16.2 | 18.4 | 38.6 |
| | 3D-LLM (BLIP2-flant5) | 39.0 | 27.8 | 21.2 | 16.6 | 18.9 | 39.3 |
| | 3D-LLM (SAM, BLIP2-t5) | **40.5** | **29.4** | **23.9** | **21.4** | **19.6** | **40.8** |
| | flant5 | 25.5 | 21.1 | 16.7 | 6.0 | 13.9 | 28.4 |
| | flamingo-SingleImage | 31.4 | 23.0 | 18.8 | 7.1 | 15.6 | 30.6 |
| | flamingo-MultiView | 33.1 | 24.7 | 21.4 | 7.3 | 16.1 | 33.2 |
| | BLIP2-flant5-SingleImage | 32.2 | 25.3 | 18.2 | 6.9 | 15.0 | 31.0 |
| Task Decomposition | BLIP2-flant5-MultiView | 33.1 | 27.0 | 20.6 | 6.9 | 15.5 | 34.0 |
| | 3D-LLM (flamingo) | 32.9 | 25.6 | 20.2 | 6.4 | 16.0 | 33.5 |
| | 3D-LLM (BLIP2-opt) | **34.1** | 27.7 | **20.8** | **7.6** | **16.5** | 35.4 |
| | 3D-LLM (BLIP2-flant5) | 33.9 | **28.1** | 20.7 | 7.4 | 15.9 | 37.8 |
| | 3D-LLM (SAM, BLIP2-t5) | 31.6 | 22.3 | 17.2 | 8.8 | 14.0 | **38.3** |

Table 4: Experimental Results on Held-In Datasets. 3D-LLMs outperform 2D VLMs.

## 5.2 More Extensive Evaluation

**Held-In Evaluation** We carry out experiments on held-in datasets of three tasks: 3D captioning, 3D-assited dialog and task decomposition. The baselines include 2D VLMs as for the held-out evaluation. We add one language-only baseline: FlanT5, which examines LLMs' ability to complete these tasks without any visual input. To evaluate the quality of responses, we include BLEU, ROUGE-L, METEOR, CIDEr as our metrics. We report the held-in evaluation performances in Table 4. From the table, we could see that 3D-LLMs could generate high-quality responses, outperforming both 2D VLMs and language-only LLMs.

**Qualitative Examples** In Figure 4, we show qualitative examples of 3D-LLM's predictions. We can see that our 3D-LLM is able to perform a variety of tasks.

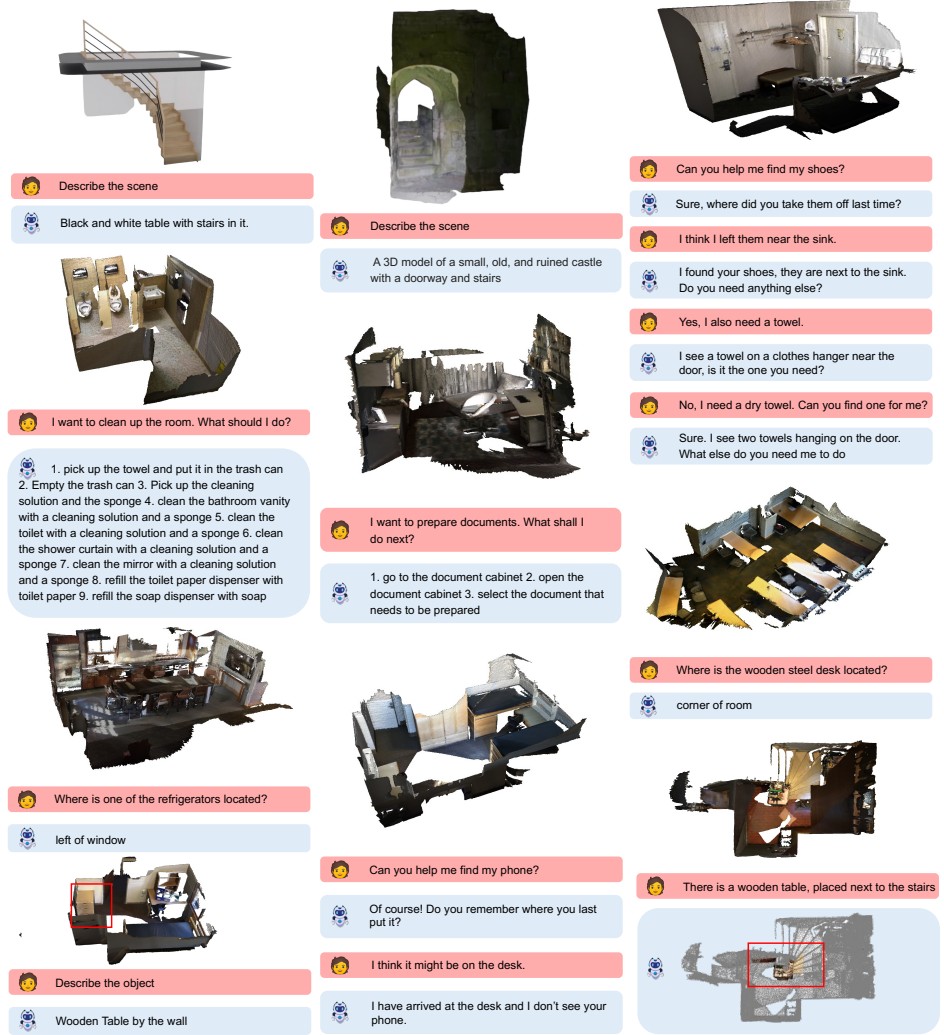

Figure 4: Qualitative examples of 3D-LLM's prediction.

## 6 Conclusion

In this paper, we propose a new family of 3D-LLMs that can take 3D representations as inputs and generate responses. We introduce a series of 3D-language data generation pipelines to generate a dataset of 1M 3D-language pairs to train our 3D-LLMs. Our 3D-LLMs leverage 2D pretrained VLMs as backbones and a novel 3D localization mechanism. Experiments show that our 3D-LLMs outperform state-of-the-art baseline models on ScanQA datasets, and could perform a diverse set of 3D-related tasks. A limitation is that the 3D feature extractor relies on 2D multi-view images, and thus all 3D scenes need to be rendered so that they can be trained in 3D-LLMs, which introduces an additional rendering process.

# 7 Acknowledgements

This work was supported by the MIT-IBM Watson AI Lab, DARPA MCS, DSO grant DSOCO21072, and gift funding from MERL, Cisco, Sony, and Amazon. We would also like to thank the computation support from AiMOS, a server cluster for the IBM Research AI Hardware Center.

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
