# A  3D-Language Data

## A.1  More details on box-demonstration-instruction based prompting

As described in the paper, we use the boxes-demonstration-instruction method for generating task decomposition data, 3D-assisted dialog data, and navigation data. In figure 5, 6 and 7, we show the prompts for generating these data, respectively. Specifically, for each *sample* in the *fewshot_samples*, the "*bbox*" is a list of the bounding boxes of a scene, and the "*response*" refers to human-written responses for demonstration. For *bbox_new_scene*, it is a list of the bounding boxes of the scene that we query the ChatGPT to give us responses.

```
messages=[{"role": "system" , "content": "You are an AI visual assistant that can analyze
a 3D scene. All object instances in this 3D scene are given, along with their center point position. The center
points are represented by a 3D coordinate (x, y, z) with units of meters. Using the provided object instance
information, design a high-level task that can be performed in this 3D scene. Besides, decomposing this high-
level task into a sequence of action steps that can be performed using the instances in this 3D scene. The
number of action steps should be LESS THAN TEN (< 10). \n\n Remenber, the high-level task and action
steps must be able to be performed in the 3D scene using the given object instances. Do not include objects
that do not exist. Do not generate similar action steps. Do not include specific locations, numbers in the action
steps."}]

for sample in fewshot_samples:
    messages.append({"role": "user", "content": '\n'.join(sample['bbox'])})
    messages.append({"role": "assistant", "content": sample['response']})

messages.append({"role": "user", "content": '\n'.join(bbox_new_scene)})
```

Figure 5: Prompts on generating task decomposition data using boxes-demonstration-instruction method.

```
messages=[{"role": "system" , "content": "You are a conversation generator in a room. All
object instances in this room are given, along with their center point position. The center points are represent
by a 3D coordinate (x, y, z) with units of meters. You need to generate 4~10 round conversation between a
human and a robot assistant. \n\n The human know all information in this room, including all objects
described above and all small things that are not visible now. The human will ask the robot to do a high-level
task. The robot will tell its observation and its state (e.g., location) to the human and will ask for help when it
is ambiguous about the task. Remember, the high-level task should be done in this room. Do not include
objects that do not exist."}]

for sample in fewshot_samples:
    messages.append({"role": "user", "content": '\n'.join(sample['bbox'])})
    messages.append({"role": "assistant", "content": sample['response']})

messages.append({"role": "user", "content": '\n'.join(bbox_new_scene)})
```

Figure 6: Prompts on generating 3D-assisted dialog data using boxes-demonstration-instruction method.

```
messages=[{"role": "system" , "content": "You are a captioner in a room. All object
instances in this room are given, along with their center point position. The center points are represent by a 3D
coordinate (x, y, z) with units of meters. For each object in the room, you need to describe it using their
affordances or functionality (how you can use it). You should write your sentence like "Where can I <do
something>" or "Find me something that I can <do something> with" or "Where should I go for
<something>"."}]

for sample in fewshot_samples:
    messages.append({"role": "user", "content": '\n'.join(sample['bbox'])})
    messages.append({"role": "assistant", "content": sample['response']})

messages.append({"role": "user", "content": '\n'.join(bbox_new_scene)})
```

Figure 7: Prompts on generating navigation data using boxes-demonstration-instruction method.

## A.2 More details on ChatCaptioner based prompting

As for the 3D models that do not have AABB bounding box annotations, we use ChatCaptioner based prompting to generate their corresponding 3D caption. We use the 3D models in the large-scale Objaverse [15] dataset. For a 3D model, we captured images from four different angles around the model, with the model as the center, covering the front, back, left, and right directions. We observed that capturing images directly from the front angle can lead to ambiguity (such as only seeing one side of the tablet). Therefore, when capturing images, we added a 15-degree pitch angle and rotated clockwise by 45 degrees on top of the original front, back, left, and right directions. For each sampled image, we use the prompt in [54] for ChatGPT to generate a question, which is fed to BLIP-2 to get the answer according to the image content. Following [54], the first question is set to "*Descripe the 3D object in the photo*". The number of conversation rounds is set to 3. We observe that the large number of conversation rounds would bring many hallucination content. We then feed the conversation history for ChatGPT to get the caption of this image. More details about prompt design can be reffed to [54].

Except for sampling images around the 3D model, for a large scene, we can also set the camera in the center of the 3D model and rotate it horizontally to sample 4 images. Since the most 3D models in Objaverse dataset are single objects, we use the former solution to sample images.

After getting caption for each image, we use the prompt shown in figure 8 for summarizing the content in all images into a 3D scene caption.

```
messages=[{"role": "assistant" , "content": "You are a 3D object descriptor. Given four
different descriptions of the same object from different viewpoints, summarize a concrete description in several
sentences. Avoid uncertain or negative information. Avoid describing the background. \n\n\ The four
descriptions are as follows. {descriptions}. Concrete 3D object description:"}]
```

Figure 8: Prompts on summarizing content in several views into a 3D scene caption on ChatCaptioner based method.

## A.3 More details on Revision based Prompting

We use the revision based prompting method to transform the information of a 3D scene into question-answering format. Specifically, we use the prompt shown in figure 9, where *context* is the generated caption of a 3D model in objaverse dataset.

```
messages=[{"role": "system" , "content": "You are a writer and editor."}]
messages=[{"role": "user" , "content": "Given the sentence {context}, revise it to a
question-answer pair of VQA task about it. Format: Question:<generated question>; Answer:<generated
answer>"}]
```

Figure 9: Prompts on summarizing content in several views into a 3D scene caption on ChatCaptioner based method.

# B Experiments

## B.1 Implementation Details

We run the models on 32 nodes, where each node has 4 V100s. The batch size is 2 for each node. The AdamW optimizer is used, with $\beta_1 = 0.9$, $\beta_2 = 0.999$, and a weight decay of 0.05. Additionally, we apply a linear warmup of the learning rate during the initial 1K steps, increasing from $10^{-8}$ to $10^{-4}$, followed by a cosine decay with a minimum learning rate of $10^{-5}$.

3D-LLMs based on pre-trained flamingo are trained using the AdamW optimizer with global norm clipping of 1, no weight decay for the perceiver resampler and weight decay of 0.1 for the other trainable parameters. The learning rate is increased linearly from 0 to $10^{-4}$ up over the first 5000 steps then held constant for the duration of training. The model is trained on 8 A100s. The batch size is 16. We use Distributed Data Parallel (DDP) to train the model.

## B.2 Captioning Results on OpenShape

We compare our captioning model with OpenShape[34], a model trained on 3 kinds of captions on 800k Objaverse data. We evaluate on OpenShape testset using pre-trained 3D-LLMs without finetuning on their training set. We ensure o data leakage among splits. Table 5 shows the results. We also show the results on our evaluation set on Table 6

|  | B4 | B3 | B2 | B1 | METEOR | ROUGHL |
|---|---|---|---|---|---|---|
| OpenShape | 1.8 | 3.6 | 8.4 | 19.7 | 5.9 | 18.4 |
| 3DLLM(BLIP2opt) | 8.5 | 11.0 | 15.2 | 21.7 | 10.3 | 29.4 |
| 3DLLM(BLIP2t5) | 9.0 | 11.4 | 16.7 | 23.6 | 11.0 | 31.3 |

Table 5: Results on OpenShape's evaluation set

| Models | B-4 | B-3 | B-2 | B-1 | METEOR | ROUGH-L |
|---|---|---|---|---|---|---|
| OpenShape | 0.9 | 2.0 | 4.8 | 11.7 | 4.3 | 14.9 |
| 3D-LLM (BLIP2-opt) | 15.9 | 20.3 | 26.7 | 35.7 | 18.7 | 40.1 |
| 3D-LLM (BLIP2-t5) | 20.1 | 24.7 | 31.0 | 39.8 | 17.7 | 42.6 |

Table 6: Results on our evaluation set

## B.3 3D Grounding

In order to examine 3D-LLMs' 3D localization abilities, we carry out a held-out experiment on ScanRefer benchmark. Specifically, ScanRefer benchmark requires the models to output object locations given a referring sentence of the objects. We finetune 3D-LLMs on ScanRefer training set and report the results on ScanRefer validation sets.

In Table 7, we show the results on. As we can see, our 3D-LLMs can have decent performances on grounding and referring, and outperform most of the baselines, showing that 3D-LLMs have the ability of 3D localization. Notably, the baseline models use ground-truth bounding boxes, or a pre-trained object detector to propose bounding boxes and classes for object proposals. Then, they use scoring modules to vote for the most likely candidate. Our method does not use any explicit object proposal module or ground truth bounding boxes, but outputs the locations of the bounding boxes directly using LLM losses for predicting tokens, while still outperforming most of the baselines. We could also see from the Avg. Dist metric the bounding boxes we predict is very close to the ground-truth brounding boxes. In Figure 10, we show some grounding results.

|  | OCRand | Vote+Rand | SCRC | One-stage | ScanRefer | 3D-LLM (flamingo) | 3D-LLM (BLIP2-opt) | 3D-LLM (BLIP2-flant5) |
|---|---|---|---|---|---|---|---|---|
| ACC@0.25 | 29.9 | 10.0 | 18.7 | 20.4 | 41.2 | 21.2 | 29.6 | 30.3 |

Table 7: Experimental Results on ScanRefer

## B.4 Object Navigation

We show the ability of our 3D-LLM to progressively understand the environment and navigate to a target object. We formulate the navigation process as a conversation. At each time step, we online build a 3D feature from the partially observed scene. We feed this feature, current agent location, and history location to the 3D-LLM for predicting a 3D waypoint the agent should go for. We then use an off-the-shelf local policy [49] to determine a low-level action (*e.g.*, go forward, turn left or right) for navigating to the waypoint. The 3D-LLM predicts "stop" if it believes the agent has reached the target object. We finetune our 3D-LLM on the object navigation episodes built on Habitat-ObjNav dataset [4].

In Figure 11, we visualize a conversation process and its corresponding navigation trajectory. At the beginning when the target object is not observed, the 3D-LLM predicts a waypoint that leads the agent to explore the area most likely containing the target object. When the agent observes the target object (*i.e.*, red box in the partially observed scene), the 3D-LLM predicts a waypoint leading the agent to it. The example episode is performed on the HM3D dataset [42] using Habitat simulator [44].

it is a dark colored two seater futon located by the door.
it is located underneath a whiteboard.

the table is white. it contains two computer monitors.

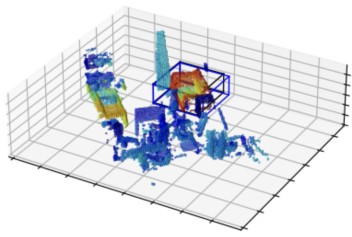 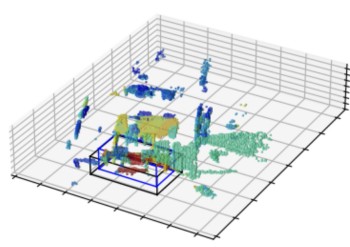

the bed is in the back left corner. it is against a window.

this is a oven stove. its white in color. its located in
between the two cherry wood marble cabinets.

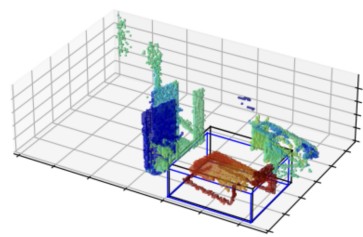 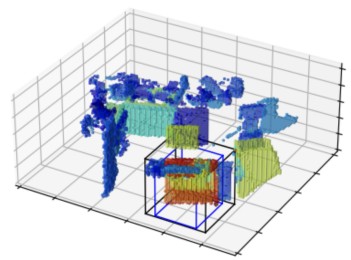

Figure 10: Qualitative examples of grounding. Black box denotes ground truth and blue box denotes predicted results. We also show the heatmap by calculating the similarity between 3D features and language features.

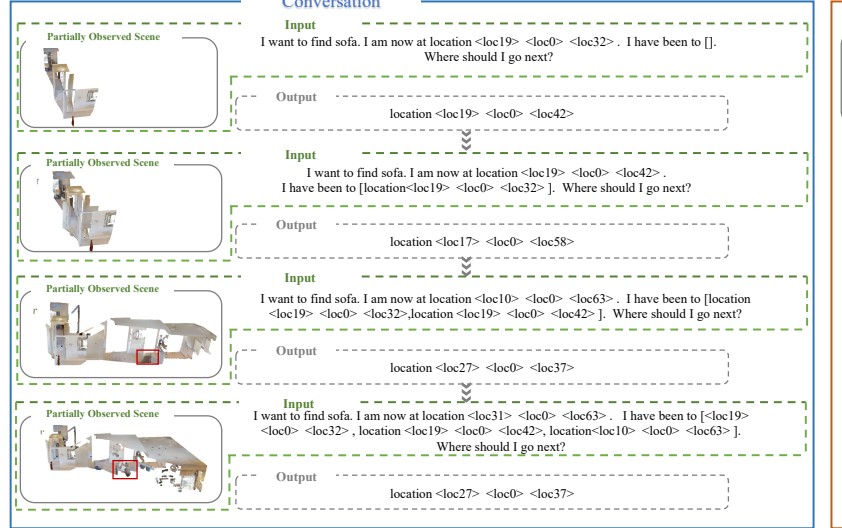 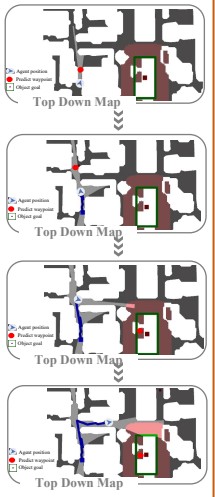

Figure 11: **Visualization of an object navigation episode.**

## B.5 Held-In Evaluation

### B.5.1 3D Dense Captioning

In Table 8, we show the results of 3D dense captioning. Specifically, given a 3D bounding box, models are expected to output the caption describing what's in that region. We can see that our 3D-LLMs outperform image-based baselines.

|  | BLEU-1 | BLEU-2 | BLEU-3 | BLEU4 | METEOR | ROUGH-L |
|---|---|---|---|---|---|---|
| flamingo-SingleImage | 21.5 | 10.5 | 6.9 | 4.1 | 11.1 | 23.4 |
| flamingo-MultiView | 24.4 | 12.3 | 7.1 | 4.6 | 11.9 | 25.8 |
| BLIP-SingleImage | 23.0 | 11.7 | 7.7 | 4.6 | 11.3 | 23.8 |
| BLIP-MultiView | 25.3 | 14.1 | 9.0 | 5.6 | 12.5 | 24.9 |
| 3D-LLM (flamingo) | 29.6 | 16.8 | 10.6 | 5.9 | 11.4 | 29.9 |
| 3D-LLM (BLIP2-opt) | 32.5 | 18.7 | 11.9 | 6.5 | 11.7 | 31.5 |
| 3D-LLM (BLIP2-flant5) | 34.3 | 20.5 | 13.2 | 8.1 | 13.1 | 33.2 |

Table 8: Experimental Results on Held-In 3D Dense Captioning Dataset.

## B.6 More Ablative Studies

### B.6.1 Ablative Studies on Flamingo Perceiver

We first examine how the perceiver resampler of Flamingo benefits the training. We carry out an ablative experiment where we take out the perceiver of the flamingo model. Table 9 shows the results. From the table, we can see that the perceiver module is indeed beneficial for the training of 3D-LLM.

|  | BLEU-1 | BLEU-2 | BLEU-3 | BLEU-4 | METEOR | ROUGH_L | CIDER | EM |
|---|---|---|---|---|---|---|---|---|
| wo/ perceiver | 29.2 | 17.2 | 11.2 | 7.4 | 11.4 | 30.4 | 58.9 | 20.6 |
| w/ perceiver | 30.3 | 17.8 | 12 | 7.2 | 12.2 | 32.3 | 59.2 | 20.4 |

Table 9: Ablative Study on the Perceiver of Flamingo Model.

## B.7 More Qualitative Examples

We show more qualitative examples in Figure 12, 13, 14.

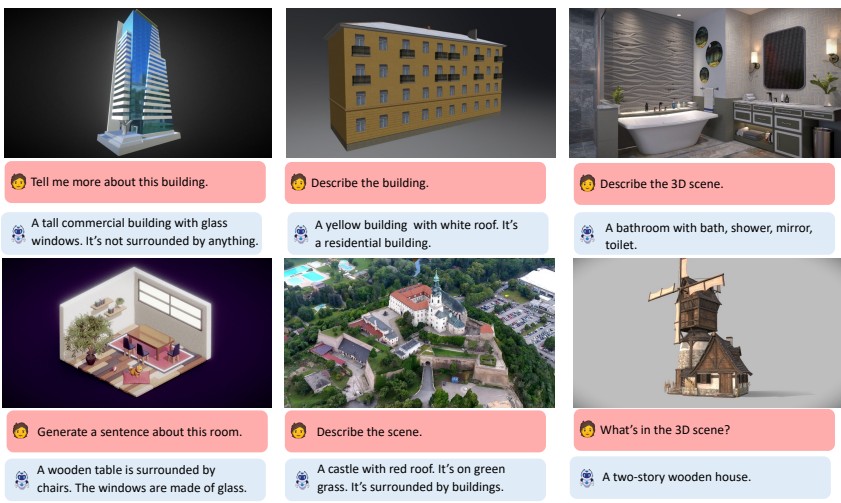

Figure 12: Qualitative Examples on 3D Captioning

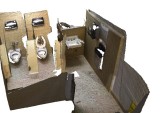

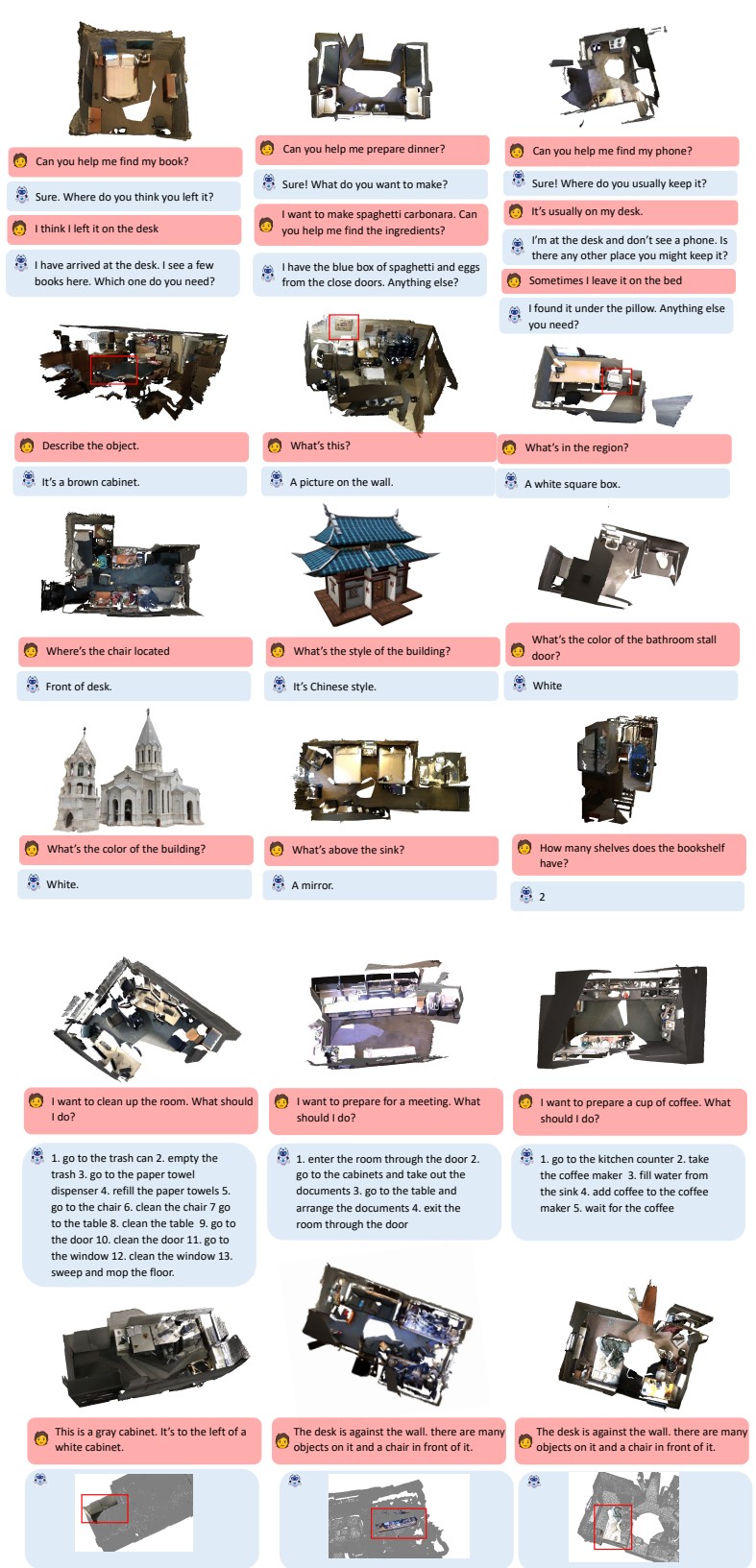

Figure 13: Qualitative Examples on 3D-Assisted Dialog, 3D Dense Captioning and Question Answering.

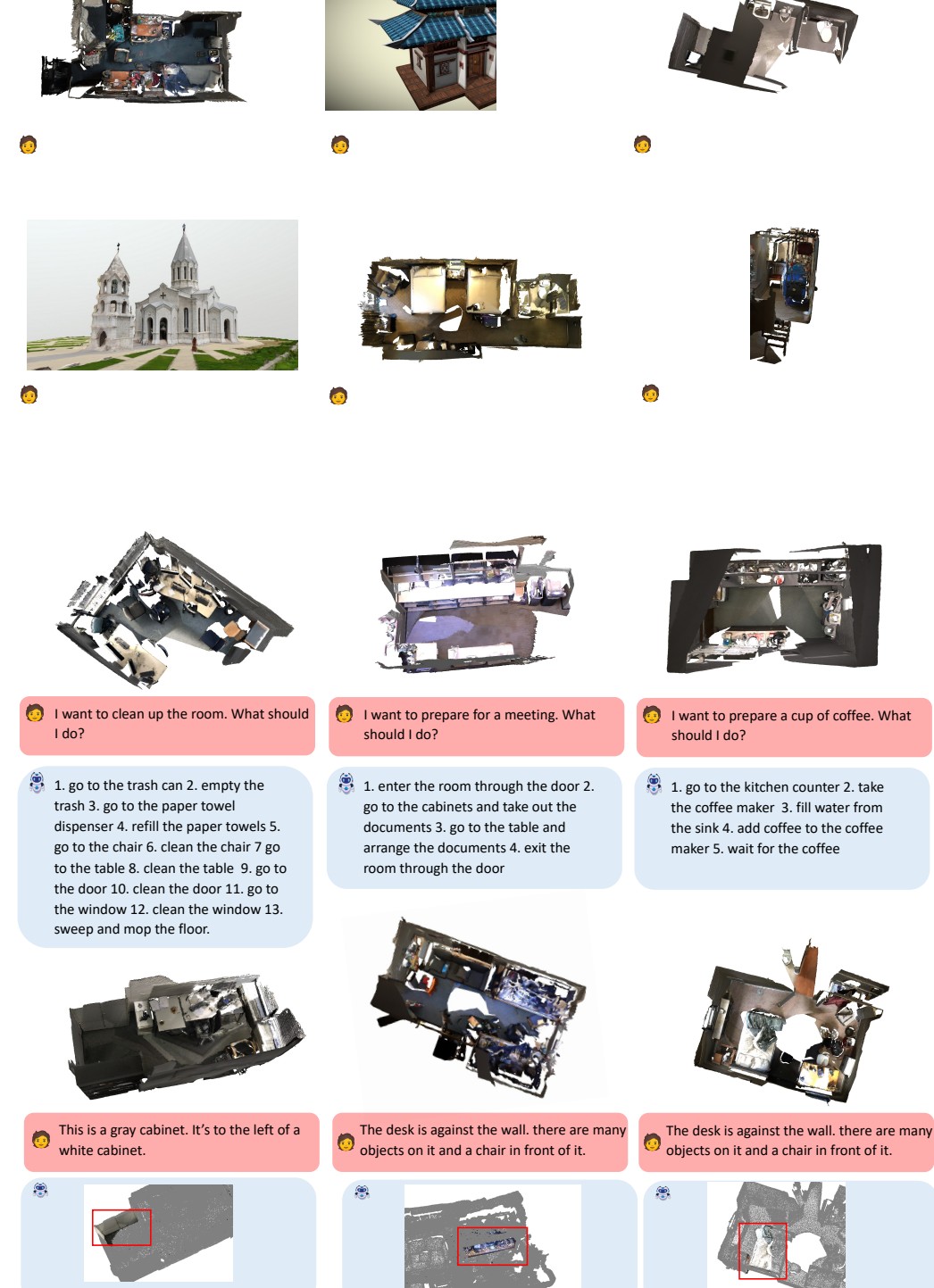

Figure 14: Qualitative Examples on Task Decomposition and Grounding (Referring).