# OpenReview forum: "3D-LLM: Injecting the 3D World into Large Language Models"
_NeurIPS.cc/2023/Conference — NeurIPS 2023 spotlight_

### Official Review · Reviewer_QPPK · 2023-06-18

**Soundness:** 2 fair
**Presentation:** 3 good
**Contribution:** 3 good
**Rating:** 7
**Confidence:** 3

**Summary:**

Due to the limited perception of 3D space by LLMs and VLMs, this paper proposes 3D-LLMs to understand spatial relationships, affordances, physics, and layout in 3D scenes. The authors generate 300K 3D-language pairs to train the 3D-LLMs, which enable better performance on various 3D understanding tasks.

**Strengths:**

1. The motivation is reasonable. The idea is clear.
2. The experimental results cover a wide range of tasks, including 3D captioning, 3D question answering, task decomposition, 3D grounding, 3D-assisted dialog, and navigation. The results are substantial and solid.
3. The organization and writing of the paper are fluent.

**Weaknesses:**

1. Is there a potential issue of data leakage? For instance, when generating the 3D-language pairs using ScanNet, the authors utilized object semantic information and bounding boxes, which might benefit the model's performance on downstream tasks such as ScanRefer and ScanQA.
2. The baselines, such as VoteNet+MCAN and ScanRefer+MCAN, should also be trained using the generated data to ensure fairness.

**Questions:**

1. Which 3D feature extractors were used for Objaverse, ScanNet, and HM3D, respectively? How many pairs were extracted for each?
2. Since the LLMs and VLMs have strong few-shot capabilities, how do the 3D-LLMs perform in zero/few-shot scenarios on held-out datasets?
3. As mentioned in the weaknesses, what results would be obtained if other baselines were also trained using the generated data?

**Limitations:**

It would be better to present the performance of 3D-LLMs in zero/few-shot scenarios on held-out datasets.

---

> ### Author Rebuttal · Authors · 2023-08-10
>
> &nbsp;
>
> *Thank you for your insightful and constructive comments! We have added additional experiments and modified our paper according to your comments.*
>
> &nbsp;
>
> > **Q1: Which 3D feature extractors were used for Objaverse, ScanNet, and HM3D, respectively? How many pairs were extracted for each?**
>
> * For Objaverse, since we render images using Blender, which gives us correct depths and camera poses, we use *direct reconstruction* to reconstruct Objaverse scenes.
> * For ScanNet, we use the original images they took to construct the scans. Since camera perspective distortion is inevitable for sensors used to take real-world pictures, we cannot use direct reconstruction to reconstruct the 3D scenes (in fact, we find that no two adjacent partial point clouds could align using direct reconstruction). Thus, we use *feature fusion with SLAM* to build the features for ScanNet.
> * For HM3D, we use mixed ways. For 3DMV-VQA [1], they do not give the depths for multi-view images. We thus use their released codes that build the features using *neural field*. However, we also use habitat-lab to collect the data of more scenes to train diverse tasks, and we are able to use *direct reconstruction* to reconstruct the data generated by habitat-lab. For example, for navigation data we need to reconstruct the partial point cloud at each observation step, and thus *direct reconstruction* is faster than *neural field*.
>
> &nbsp;
>
> > **Q2: Since the LLMs and VLMs have strong few-shot capabilities, how do the 3D-LLMs perform in zero/few-shot scenarios on held-out datasets?**
>
> Thank you for suggesting! We attach the results of the pre-trained models below.
>
> &nbsp;
>
> **Table A. Zero-shot Performances of Pretrained Models, on ScanQA.**
> || B-1  | B-2  | B-3  | B-4  | METEOR | ROUHE-L | CIDER | EM   |
> |-|-|-|-|-|-|-|-|-|
> | VoteNet+MCAN*  |28.0 | 16.7 | 10.8 | 6.2  | 11.4   | 29.8    | 54.7  | 17.3 |
> | ScanRefer+MCAN* |26.9 | 16.6 | 11.6 | 7.9  | 11.5   | 30      | 55.4  | 18.6 |
> | ScanQA* | 30.2 | 20.4 |15.1 | 10.1 | 13.1   | 33.3    | 64.9  | 21.0 |
> | LLaVA(zero-shot)| 7.1 | 2.6  | 0.9  | 0.3  | 10.5   | 12.3    | 5.7   | 0.0  |
> | **3D-LLM (flamingo) - PT** |  22.5  |  12.1   |   5.5   |    2.7    |   10.0    |   24.3   |  49.8    | 14.0 |
> | **3D-LLM (BLIP2-opt) - PT**  | 26.4 | 14.3 | 7.2  | 3.4  | 11.9   | 27.1    | 52.7   | 13.2|
> | **3D-LLM (BLIP2-flant5) - PT** |  28.6 | 17.0 | 9.9  | 6.6  | 12.3   | 28.0    | 52.7   |13.8 |
> | 3D-LLM (flamingo) |  30.3 | 17.8 | 12.0 | 7.2  | 12.2   | 32.3    | 59.2  | 20.4  |
> | 3D-LLM (BLIP2-opt) |  35.9 | 22.5 | 16.0 | 9.4  | 13.8   | 34.0    | 63.8   | 19.3   |
> | 3D-LLM (BLIP2-flant5) |  39.4 | 25.2 | 18.3 | 12.3 | 14.9   | 35.9    | 69.3   | 20.5 |
>
> &nbsp;
>
> **Table B. Zero-shot Performances of Pretrained Models, on 3DMV-VQA.**
>  Methods | Concept | Counting | Relation | Comparison | Overall
> ---|---|---|---|---|---
>  CNN+LSTM | 57.8 | 22.1 | 35.2 | 59.7 | 37.8
>  MAC | 62.4 | 19.7 | 47.8 | 62.3 | 46.7
>  MAC(V) | 60.0 | 24.6 | 51.6 | 65.9 | 50.0
>  NS-VQA | 59.8 | 21.5 | 33.4 | 61.6 | 38.0
>  ALPRO | 65.8 | 12.7 | 42.2 | 68.2 | 43.3
>  LGCN | 56.2 | 19.5 | 35.5 | 66.7 | 39.1
>  3D-Feature+LSTM | 61.2 | 22.4 | 49.9| 61.3 | 48.2
>  3D-CLR (Ours) | 66.1 | 41.3 | 57.6 | 72.6 | 57.7
>  **3D-LLM (flamingo) - PT** | 60.5 | 18.9 | 47.6 | 63.7 | 45.7
>  **3D-LLM (BLIP5-opt) - PT** |60.1|17.8 | 44.2 | 58.9 | 42.9
>  **3D-LLM (BLIP2-flanT5) - PT** | 61.0 | 18.4 | 45.0 | 62.0 | 43.9
>  3D-LLM (flamingo) | 68.9 | 32.4 | 61.6 | 68.3 | 58.6
>  3D-LLM (BLIP5-opt) | 63.4 | 30.7 | 57.6 | 65.2 | 54.9
>  3D-LLM (BLIP2-flanT5) | 68.1 | 31.4 | 55.1 | 69.7 | 54.6
>
> &nbsp;
>
> **Table C. Zero-shot Performances of Pretrained Models, on ScanRefer.**
> ||Acc@0.25
> -|-
> OracleRand|29.9
> OracleRefer|40.6
> VoteNetRand|10.0
> SCRC|18.7
> OneStage|20.4
> VoteNetGRU|39.5
> ScanRefer|41.2
> **3DLLM(BLIP2t5)-PT**|20.1
> **3DLLM(BLIP2t5)-new-PT**|26.7
> 3DLLM(BLIP2t5)|30.3
> 3DLLM(BLIP2t5)-new|35.2
>
> *"new" means the new model we train after submission*
>
> &nbsp;
>
> > **Q3: what results would be obtained if other baselines were also trained using the generated data?**
>
> * Thank you for suggesting! However, these baselines rely on privileged information that is not available in our current pre-training dataset. The training of VoteNet and ScanRefer requires the ground-truth segmentations of the objects in the scenes. However, we do not have these annotations in the Objaverse dataset. We do think such information is crucial and one future step for us is to label the segmentations of the Objaverse dataset to help improve the performances of 3D-LLMs.
> * The ScanQA baseline uses an answer classification module in addition to object localization, detection and object classification modules to propose answers. Thus, it does not have the language generation ability which is essential for our tasks (*e.g.*, captioning and dialogue).
> * Despite these limitations, we still provide the results of VoteNet+MCAN and ScanRefer+MCAN below, where we use pre-trained VoteNet and ScanRefer and train the MCAN part (the results might be not so meaningful since there is non-trivial domain gap between the data VoteNet and ScanRefer trained on and our pre-training data). We pre-train the models on our pre-training dataset and finetune them on ScanQA.
>
> &nbsp;
>
> **Table D. Performances on ScanQA baselines when pretrained on 3D-language data.**
> | | B-1  | B-2  | B-3  | B-4  | METEOR | ROUHE-L | CIDER | EM   |
> |---|------|------|------|------|--|-|-|------|
> | VoteNet+MCAN*  | 28.0 | 16.7 | 10.8 | 6.2  | 11.4   | 29.8    | 54.7  | 17.3 |
> | VoteNet+MCAN*  (pretrained)| 26.4 | 15.4 | 9.1 |5.8  |10.4  | 25.3    | 50.6  | 15.9 |
> | ScanRefer+MCAN* |26.9|16.6| 11.6| 7.9 | 11.5| 30| 55.4  | 18.6 |
> | ScanRefer+MCAN* (pretrained)  | 28.3 |17.0|12.1|7.6|11.2| 29.7| 52.6  | 16.9 |
>
> &nbsp;
>
> [1] 3D Concept Learning and Reasoning from Multi-View Images. Hong et al. 2023
>
> &nbsp;
>
> *We sincerely appreciate your comments. Please feel free to let us know if you have further questions.*
>
> &nbsp;
>
> Best,
> Authors

---

> > ### Comment · Reviewer_QPPK · 2023-08-14
> >
> > The response clearly solves my concerns. Thus, I improve my final rating from 6 to 7.

---

### Official Review · Reviewer_6ag4 · 2023-07-06

**Soundness:** 3 good
**Presentation:** 3 good
**Contribution:** 3 good
**Rating:** 6
**Confidence:** 4

**Summary:**

This paper proposes a new family of 3D-LLMs that can take 3D representations as inputs and generate responses, it introduces a series of 3D-language data generation pipelines to generate a dataset of 300K 3D-Language pairs from different tasks for the training.

**Strengths:**

The proposed approach seems to be valid and functioning, and the authors say they plan to release the 3D-Language dataset as well.
Overall it demonstrates how to use 2d features to gather 3D features and then inject them into LLM.

**Weaknesses:**

The weakness of this paper is on how to gather the 3D representation, current pipeline seems to be relying on 2D multi-view images, which will introduce extra complexity/latency and limitations. Also, when you project 3D data to a series of multi-view images, it's likely that there will be some information loss, even though the reviewer does appreciate that the authors do realize this issue and come up with some remedy approaches like the "3D Localization Mechanism", it seems functioning to some extent, but it might not be the optimal approach eventually.


**Questions:**

Typo: there is an extra "." in line 132.
The reviewer is curious whether the authors have tried to train it directly using the language-3D dataset for the 3D encoder directly without leveraging the images as the bridge, and how will it perform differently.
In response to that suggestion, the reviewer would like to mention some related works that have explored the alignment of 3D-image-language triplets and consequently trained the 3D encoder to have language context [1][2]. Given the abundance of 3D object data, employing a pre-trained 3D encoder, and subsequently fine-tuning it with the LLM in this case could be a potentially promising strategy, and it might yield more fruitful results compared to training a model from scratch using the same data the authors have collected.
[1]: "ULIP: Learning a Unified Representation of Language, Images, and Point Clouds for 3D Understanding" -- CVPR2023
[2]: "CLIP2: Contrastive Language-Image-Point Pretraining from Real-World Point Cloud Data" -- CVPR2023


**Limitations:**

refer to the weaknesses section

---

> ### Author Rebuttal · Authors · 2023-08-10
>
> &nbsp;
>
>
> *We appreciate the positive and insightful comments from you! We address your concerns in detail below.*
>
> &nbsp;
>
> > **Q1: The reviewer is curious whether the authors have tried to train it directly using the language-3D dataset for the 3D encoder directly without leveraging the images as the bridge, and how will it perform differently. In response to that suggestion, the reviewer would like to mention some related works that have explored the alignment of 3D-image-language triplets and consequently trained the 3D encoder to have language context. Given the abundance of 3D object data, employing a pre-trained 3D encoder, and subsequently fine-tuning it with the LLM in this case could be a potentially promising strategy, and it might yield more fruitful results compared to training a model from scratch using the same data the authors have collected.**
>
> Thank you for suggesting! The suggestion will be immensely helpful in enhancing the paper’s quality and assisting the readers in understanding the paper's contributions.
>
> We replace 3D-LLMs' features with features from pretrained 3D encoders (ULIP) [1]. The results shown in Table A and B suggest that LLMs with 3D encoders have very poor performances, inferior to 3D-LLMs by a large margin.
>
> &nbsp;
>
> **Table A. Experimental Results of Pretrained 3D Encoder with LLMs, on ScanQA.**
> || EM|B-1|B-2|B-3|B-4|METEOR|ROUHE-L|CIDER|
> |-|-|-|-|-|-|-|-|-|
> |VoteNet+MCAN*|17.3|28.0|16.7|10.8|6.2|11.4|29.8|54.7|
> |ScanRefer+MCAN*|18.6|26.9|16.6|11.6|7.9|11.5|30|55.4|
> |ScanQA*|21.0|30.2|20.4|15.1|10.1|13.1|33.3|64.9|
> |flamingo-SingleImage|16.9|23.8|14.5|9.2|8.5|10.7|29.6|52.0|
> |flamingo-MultiView|18.8|25.6|15.2|9.2|8.4|11.3|31.1|55.0|
> |BLIP2-flant5-SingleImage|13.3|28.6|15.1|9.0|5.1|10.6|25.8|42.6|
> |BLIP2-flant5-MultiView|13.6|29.7|16.2|9.8|5.9|11.3|26.6|45.7|
> |**ULIP_PointMLP+flant5**|7.5|11.0|18.4|7.2|2.7|1.4|7.4|18.1|26.9|
> |**ULIP_PointMLP+opt**|8.4|19.1|7.3|2.7|1.9|7.4|18.2|28.0|
> |**ULIP_PointBERT+flant5**|14.5|29.2|17.9|10.3|6.1|11.6|28.1|50.9|
> |**ULIP_PointBERT+opt**|13.8|28.8|16.9|9.7|5.9|11.3|27.9|50.5|
> |3D-LLM (flamingo)|20.4|30.3|17.8|12.0|7.2|12.2|32.3|59.2|
> |3D-LLM (BLIP2-opt)|19.3|35.9|22.5|16.0|9.4|13.8|34.0|63.8|
> |3D-LLM (BLIP2-flant5)|20.5|39.4|25.2|18.3|12.3|14.9|35.9|69.3|
>
> &nbsp;
>
> **Table B. Experimental Results of Pretrained 3D Encoder with LLMs, on Captioning.**
> |Models|BLEU-1|BLEU-2|BLEU-3|BLEU-4|METEOR|ROUGH-L|
> |-|-|-|-|-|-|-|
> |**ULIP-PointMLP+flant5**|24.6|20.8|14.9|10.4|12.5|35.3|
> |**ULIP-PointMLP+opt**|24.4|20.4|14.3|10.8|12.1|34.1|
> |**ULIP-PointBERT+flant5**|26.0|21.9|15.7|10.0|14.8|33.8|
> |**ULIP-PointBERT+opt**|24.3|20.7|15.8|11.3|12.1|39.5|
> |3D-LLM (flamingo)|36.2|24.8|19.0|16.0|17.6|40.8|
> |3D-LLM (BLIP2-opt)|35.7|26.7|20.3|15.9|18.7|40.1|
> |3D-LLM (BLIP2-t5)|39.8|31.0|24.7|20.1|17.7|42.6|
>
> &nbsp;
>
>
> > **Q2: The weakness of this paper is on how to gather the 3D representation, current pipeline seems to be relying on 2D multi-view images, which will introduce extra complexity/latency and limitations. Also, when you project 3D data to a series of multi-view images, it's likely that there will be some information loss, even though the reviewer does appreciate that the authors do realize this issue and come up with some remedy approaches like the "3D Localization Mechanism", it seems functioning to some extent, but it might not be the optimal approach eventually.**
>
> * We admit that one limitation of the paper is that the 3D representation relies on 2D multi-view images, which may result in extra complexity and information loss. This limitation is also covered in Line 295 of our submission.
> * However, We want to emphasize that building LLMs on 3D world is extremely challenging, due to the severely limited amount of existing 3D data and extreme difficulty to gather more such data. There are two potential solutions to approach the unexplored area:
>
>     * The first one is to encode 3D features using pre-trained 3D encoders and input to LLMs, which is non-trivial due to the limited scale and diversity of 3D assets. Existing 3D encoders focus on simple objects instead of real-world scenes[1,2]. In Q1, we show that such pre-trained 3D encoders have very poor performances.
>
>     * The second one is learning models of the 3D world by leveraging the abundance of 2D multiview images and aggregating 2D features to 3D. There has been a recent surge of works using 2D features to construct 3D representations[3,4], and there is significantly more plentiful 2D multiview data of scenes than there are 3D scans of these scenes. These methods further show superior ability in zero-shot and open-vocabulary reasoning, indicating this is a promising and appropriate strategy for processing 3D scenes at the moment.
>
>    * **While we do not come to the conclusion that the second solution, utilized by 3D-LLM, is better than the first one, and indeed has several limitations, it is the most practical solution given the limited data for the time being.** 3D-LLM serves as a promising first step into exploring LLMs grounded in the 3D physical world and brings inspiration into the community. We believe that in the future, more powerful models will be built upon the combination of the two solutions. More discussions will be added in the revision.
>
> &nbsp;
>
> [1] ULIP: Learning a Unified Representation of Language, Images, and Point Clouds for 3D Understanding
>
> [2] CLIP2: Contrastive Language-Image-Point Pretraining from Real-World Point Cloud Data
>
> &nbsp;
>
>
> *We sincerely appreciate your comments. Please feel free to let us know if you have further questions. Thank you again for your time!*
>
> &nbsp;
>
> Best,
> Authors

---

> > ### Comment · Reviewer_6ag4 · 2023-08-19
> > **Thanks for the rebuttal**
> >
> > Thanks for the authors' rebuttal, the reviewer has read the rebuttal in detail and would like to maintain the positive rating.

---

> ### Author Response · Authors · 2023-08-18
> **Follow-up on rebuttal**
>
> Dear Reviewer,
>
> Thanks again for your suggestions to strengthen this work! As the rebuttal period is approaching the end soon, we want to know if our response has answered your questions and addressed your concerns. If no, we are more than happy to provide further modifications. If yes, would you kindly consider raising the score?
>
> Thanks again for your truly constructive and insightful feedback.
>
> Best,
> Authors

---

> ### Comment · Area_Chair_rcuh · 2023-08-19
>
> Dear Reviewer 6ag4,
>
> We are nearing the end of the discussion period with authors.
>
> The authors have responded in detail to your review, so pls minimally read and acknowledge their rebuttal, and state which (if any) issues you still do not find to be satisfactorily addressed.
>
> You should do so as soon as possible.
>
> Thanks, AC

---

### Official Review · Reviewer_G3Mn · 2023-07-06

**Soundness:** 4 excellent
**Presentation:** 4 excellent
**Contribution:** 4 excellent
**Rating:** 8
**Confidence:** 4

**Summary:**

This paper proposes a new framework named 3D-LLM which leverages LLM to understand the 3D world. Specifically, 3D-LLM can take 3D point clouds as inputs to conduct various 3D tasks, including captioning, dense captioning, 3D question answering, task decomposition, 3D grounding, 3D-assisted dialog, navigation, and so on. To achieve this goal, this paper designs three types of prompting mechanisms to generate over 300k 3D-language data. It proposes a 3D feature extractor that obtains 3D features from rendered multiview images and takes pretrained 2D VLMs as backbones to train 3D-LLM. Both benchmark results on held-out and held-in data show the effectiveness of the proposed framework, which achieves SOTA performance on ScanQA benchmark.

**Strengths:**

1. This paper is well written.
2. The idea of injecting the 3D world into large language models is novel.
3. The technical contributions, including data generation, overall framework, and experimental analysis are solid and convincing.
4. The proposed 3D-LLM achieves impressive results on both quantitative and qualitative results.

**Weaknesses:**

This paper is satisfactory. I only have some minor comments.
1. In Table 4, the ablation study could be more comprehensive, where the baseline should remove all of the position embedding, location tokens, and localization. The ablation study should begin with this baseline and show all possible combinations of these three designs.
2. In Figure 1, three approaches generate 3D features, what is the effectiveness of each of them? The experimental results should be included.
3. Limitations of 3D-LLM should be discussed.

**Questions:**

See weakness.

**Limitations:**

The limitations are not discussed.

---

> ### Author Rebuttal · Authors · 2023-08-10
>
> &nbsp;
>
> *We appreciate the positive and constructive comments from you, which are essential for improving the paper! We have conducted your suggested experiments. We will update all results in the paper.*
>
> &nbsp;
>
> > **Q1: In Table 4, the ablation study could be more comprehensive, where the baseline should remove all of the position embedding, location tokens, and localization. The ablation study should begin with this baseline and show all possible combinations of these three designs.**
>
> Sorry for the confusion! The localization in the table actually means position embeddings plus location tokens. As can be seen from the paper Line 202: *"It consists of two parts: 1) Augmenting 3D features with position embeddings 2) Augmenting LLM vocabularies with location tokens"*. Therefore, we think we have covered all the cases. We will modify the writing to make it more understandable here.
>
>
> &nbsp;
>
> > **Q2: In Figure 1, three approaches generate 3D features, what is the effectiveness of each of them? The experimental results should be included.**
>
> * The three approaches are meant for different kinds of data. For example, for real-world scans like ScanNet, camera perspective distortion is inevitable and thus we cannot use direct reconstruction to reconstruct the 3D scenes (in fact, we find that no two adjacent partial point clouds could align using direct reconstruction). Thus, we use *feature fusion with SLAM* to build the features for ScanNet. For 3DMV-VQA, the depth data is not released, and thus we can only use *neural field* to reconstruct the 3D scenes. On the other hand, for Objaverse data, since we are rendering with blender which gives us correct camera poses and depths, we use *direct reconstruction*.
> * To shed light on the performances of different approaches, we conduct an experiment where we use all three different features to construct the ScanNet features, with 3D-LLM (BLIP2-flant5) as our model, and show them in Table A. We can see that the *direct reconstruction* result is inferior to the two others, mainly because the scannet features we obtain via direct reconstruction have noises due to camera perspective distortion. The results of *feature fusion* and *neural field* are on par since they can both correctly reconstruct the 3D scenes. We will update more experimental results to the camera ready version concerning this question you raised.
>
>
> &nbsp;
>
> **Table A. Comparison among 3D feature generation approaches.**
> |                           | B-1           | B-2           | B-3         | B-4          | METEOR        | ROUHE-L       | CIDER         | EM            |
> |----------------------------|---------------|---------------|-------------|--------------|---------------|---------------|---------------|---------------|
> | 3D-LLM   (Direct Reconstruction)                  | 34.6 | 22.1 | 15.7        | 9.1          | 13.5 | 33.0 |55.7 | 18.9 |
> | 3D-LLM   (feature fusion)                 |  39.4 | 25.2 | 18.3 | 12.3 | 14.9   | 35.9    | 69.3   | 20.5 |
> | 3D-LLM   (Neural Field)                  | 39.1 | 25.0 | 18.5      | 12.1        | 15.2 | 36.0 | 67.3 | 20.3 |
>
>
> &nbsp;
>
> > **Q3: Limitations of 3D-LLM should be discussed.**
>
> In Line 295 of the paper, we gave one limitation of 3D-LLM: *"A limitation is that the 3D feature extractor relies on multi-view images"*.
>
> We would like to share with you some more limitations which are crucial for further improvement of this paper:
>
> * For the grounding mechanism, we input detailed text descriptions to refer to an object and train 3D-LLMs to output location tokens for these objects. However, the referring sentence might contain multiple hops of relations, and directly training on such corpus is non-trivial since the models need to simultaneously learn both semantics and relationships. A better way to improve the grounding mechanism is to assign location tokens to each noun in all data, like in [1].
> * We find that 3D-LLMs, like a lot of 2D VLMs, are bad at grounding relationships [2]. This problem is more salient for 3D tasks which have more complex spatial relationships. Relational modules need to be added to the models.
> * We do not have ego-centric or robot-centric data in our 3D-language data. Therefore, current 3D-LLMs are unable to solve embodied robotics tasks. Such data and tasks are crucial for equipping 3D-LLMs with the ability to understand the complex 3D physical world.
>
>
> &nbsp;
>
> [1] Kosmos-2: Grounding Multimodal Large Language Models to the World. Zhiliang Peng et al. 2023
>
> [2] Going Beyond Nouns With Vision & Language Models Using Synthetic Data. Paola Cascante-Bonilla et al. 2023
>
> &nbsp;
>
> *Please let us know if you have any further questions for our paper. We sincerely appreciate your time for reviewing this paper and raising the valuable suggestions! Thank you again!*
>
> &nbsp;
>
> Best,
> Authors

---

> > ### Comment · Reviewer_G3Mn · 2023-08-14
> >
> > My concerns have been addressed in the rebuttal and I am satisfied. Therefore, I will maintain my previous rating of 8.

---

> ### Comment · Reviewer_G3Mn · 2023-08-21
> **My last question Question**
>
> This work is quite intriguing. My last question is when the generated dataset, the training code, and the evaluation code will be released.
> I believe that these materials would be helpful for researchers in the community and hope they can be publicly available as soon as possible.

---

> > ### Author Response · Authors · 2023-08-21
> >
> > Dear Reviewer,
> >
> > Thank you for asking. The datasets and codes will be publicly available very soon.
> >
> > Best,
> > Authors

---

### Official Review · Reviewer_KXNR · 2023-07-07

**Soundness:** 3 good
**Presentation:** 3 good
**Contribution:** 3 good
**Rating:** 6
**Confidence:** 4

**Summary:**

In this paper, the authors tried to leverage LLM to understand the 3D scene. Specifically, the authors use both grounding and captioning/QA datasets to tune the model. Specifically, the authors adopt the three 2D to 3D feature transformation techniques to let the model have a sense of the 3D features.

**Strengths:**

1. The motivation is clear.
2. The paper is easy to read.

**Weaknesses:**

- When you aggregate the 2D features to 3D. It could be time-consuming and ill-posed.
- The performance under the pre-trained weights upon all sources of the pretraining data is missing. Authors should report such results. In this way, you will see the effect of finetuning.
- How to measure hallucination?
- In 3D captioning, authors should report the baseline methods from recent papers.
- I think this paper is more like a transition paper. What if we are given a (1) pure 3D point cloud, such as Modelnet? (2) Pointed cloud with limited images, such as KITTI? Here the ill-posed problem is that some 3D points can be never mapped into RGB pixel(s). In this sense, RGBD cameras are the only lucky sensor that can do the 2D to 3D projection.
- The results on ScanRefer are far from satisfactory. I know that this task is hard, however, the baselines are not moderate/strong enough. I do not it make sense to put the random guessing or one-stage method numbers here...
- What is the resolution of the position embeddings and location tokens? Given a large 3D scene such as LiDAR, accurately localizing the objects needs quite a lot of tokens. For example, if a scene is 50m*50m*6m, then you need a lot of tokens.
- For grounding, what we expect is <loc_x><loc_y><loc_z>, what if the outputs are not what we expected, such as <loc_x><loc_z> or <loc_x>text1<loc_y>text2<loc_z>?

**Questions:**

See comments above.

**Limitations:**

See comments above.

---

> ### Author Rebuttal · Authors · 2023-08-10
>
> We'd like to express our sincere gratitude for your thorough review of our paper. We greatly appreciate your suggestions which are crucial in improving the quality of our paper.
> > Q1: Aggregating 2D features to 3D is ill-posed
>
> Thanks for raising this concern. We want to emphasize that building LLMs on 3D world is extremely challenging, due to severely limited amount of existing 3D data and extreme difficulty to gather more such data. There are two potential solutions to approach the unexplored area:
>
> The first one is to encode 3D features using pre-trained 3D encoders and input to LLMs, which is non-trivial due to the limited scale and diversity of 3D assets. Existing 3D encoders focus on simple objects instead of real-world scenes[1,2]. Taking the suggestion of reviewer 6ag4, we replace 3D-LLM's features with features from pretrained 3D encoders (ULIP). Results below show their performances are very poor.
> ||EM|B1|B2|B3|B4|METEOR|ROUHEL|CIDER
> -|-|-|-|-|-|-|-|-
> ULIPPointMLP+t5|7.5|18.4|7.2|2.7|1.4|7.4|18.1|26.9
> ULIPPointMLP+opt|8.4|19.1|7.3|2.7|1.9|7.4|18.2|28.0
> ULIPPointBERT+t5|14.5|29.2|17.9|10.3|6.1|11.6|28.1|50.9
> ULIPPointBERT+opt|13.8|28.8|16.9|9.7|5.9|11.3|27.9|50.5
> 3DLLM(BLIP2opt)|19.3|35.9|22.5|16.0|9.4|13.8|34.0|63.8
> 3DLLM(BLIP2t5)|20.5|39.4|25.2|18.3|12.3|14.9|35.9|69.3
>
> The second one is learning models of the 3D world by leveraging the abundance of 2D multiview images and aggregating 2D features to 3D. There has been a recent surge of works using 2D features to construct 3D representations[3,4], and there is significantly more plentiful 2D multiview data of scenes than there are 3D scans of these scenes. These methods further show superior ability in zero-shot and open-vocabulary reasoning, indicating this is a promising and approariate strategy for processing 3D scenes at the moment.
>
> **While we do not come to the conclusion that the second solution, utilized by 3D-LLM, is better than the first one, and indeed has several limitations, it is the most practical solution given the limited data for the time being.** 3D-LLM serves as a promising first step into exploring LLMs grounded in the 3D physical world and brings inspiration into the community. We believe that in the future, more powerful models will be built upon the combination of the two solutions. More discussions will be added in the revision.
>
> > Q2: Performance under pre-trained weights
>
> Attached below.
>
> ||EM|B1|B2|B3|B4|METEOR|ROUHEL|CIDER
> |-|-|-|-|-|-|-|-|-
> 3DLLM(flamingo)|14.0|22.5|12.1|5.5|2.7|10.0|24.3|49.8
> 3DLLM(BLIP2opt)|13.2|26.4|14.3|7.2|3.4|11.9|27.1|52.7
> 3DLLM(BLIP2t5)|13.8|28.6|17.0|9.9|6.6|12.3|28.0|52.7
> > Q3: Hallucination
>
> We explore two metrics[5].
>
> CHAIR: Portion of hallucinated objects in all mentioned ones
>
> HRF@k: Portion of frequent objects in hallucinated ones
>
> We report scores of two tasks.
>
> |||CHAIR↓|HRF@10↓
> -|-|-|-
> Task Decom.|t5|55.6|69.7
> ||BLIP2t5-Image|35.4|56.8
> ||3DLLM(BLIP2t5)|7.5|39.2
> Dialog|t5|51.6|59.9
> ||BLIP2t5-Image|29.2|54.6
> ||3DLLM(BLIP2t5)|5.5|33.2
> > Q4: 3D captioning baseline
>
> We attach the result of OpenShape[6], a model trained on 3 kinds of captions on 800k Objaverse data. We evaluate on OpenShape testset using pre-trained 3D-LLMs without finetuning on their training set. No data leakage among splits.
>
> ||B4|B3|B2|B1|METEOR|ROUGHL
> -|-|-|-|-|-|-|
> OpenShape|1.8|3.6|8.4|19.7|5.9|18.4
> 3DLLM(BLIP2opt)|8.5|11.0|15.2|21.7|10.3|29.4
> 3DLLM(BLIP2t5)|9.0|11.4|16.7|23.6|11.0|31.3
>
> Results on our Objaverse testset.
>
> ||B4|B3|B2|B1|METEOR|ROUGHL
> -|-|-|-|-|-|-
> OpenShape|0.9|2.0|4.8|11.7|4.3|14.9
> 3DLLM(BLIP2t5)|20.1|24.7|31.0|39.8|17.7|42.6
>
> 3D-LLMs outperform OpenShape a lot, even on OpenShape's testset, with  less than 10% training data of OpenShape's. We give both models' qualitative results in PDF. We'll run more models if the reviewer has suggestions.
>
> > Q5: Some 3D points can be never mapped into pixels.
> *  We agree that our framework is not suitable for all kinds of point clouds, but note that many existing pointclouds can be mapped and rendered with 2D images to get features.
> *  In the PDF, we show the results on ModelNet with rendered images, features and 3D-LLM responses. For KITTI, we could take a partial point cloud at each step, which we already did that with our navigation task. KITTI results are also in the PDF.
> * For pointclouds that may not be easily rendered, we could learn a separate stream to directly encode 3D features from the pointclouds and align them with the 3D features from RGB images. This allows us to take advantage of both 3D data and plentiful 2D multiview data.
>
> > Q6: ScanRefer results
> * We gladly share with you our new result, outperforming previous one by 5%. We also add the required stronger baselines. New result is achieved by a few modifications: 1) Before we followed the original version of [3] using MaskFormer to get dense 2D features, now we use Segment Anything 2) Pos embeddings are added to the features rather than concatenated.
>
> ||Acc@0.25
> -|-
> OracleRand|29.9
> OracleRefer|40.6
> VoteNetRand|10.0
> SCRC|18.7
> OneStage|20.4
> VoteNetGRU|39.5
> ScanRefer|41.2
> 3DLLM(BLIP2t5)|30.3
> 3DLLM(BLIP2t5)-new|35.2
>
> > Q7: Resolution
>
> Pos embedding: $256^3$. Location token: $64^3$ (64 tokens,applied in 3 dims). We could expand the token number (e.g., 64 to 256) for larger scenes.
> > Q8: Wrong grounding output
>
> Grammarly-incorrect output is considered wrong with 0 IOU.
>
> [1]PointCLIP: Point Cloud Understanding by CLIP
>
> [2]ULIP: Learning a Unified Representation of Language, Images, and Point Clouds for 3D Understanding
>
> [3]ConceptFusion: Open-set Multimodal 3D Mapping
>
> [4]3D Concept Learning and Reasoning from Multi-View Images
>
> [5]Evaluating Object Hallucination in Large Vision-Language Models
>
> [6]OpenShape: Scaling Up 3D Shape Representation Towards Open-World Understanding
>
> We wish that our response has addressed your concerns and turns your assessment to the positive side. If you have more questions, feel free to let us know during the rebuttal window. Thank you!

---

> ### Author Response · Authors · 2023-08-17
> **Follow-up on rebuttal**
>
> Thank you for your comments! We would like to follow up on whether our response and our additional experiments on pre-trained models, ULIP, hallucination and captioning baselines have cleared your concerns. We are looking forward to your further comments on these perspectives, and we are more than happy to make further adjustments if necessary!
>
> Thanks for your time again!

---

> > ### Comment · Reviewer_KXNR · 2023-08-17
> >
> > I really appreciate the authors for producing such a great number of new experiments during the rebuttal. Most of my concerns are addressed. Though I am concerned with a true 3D point cloud encoder, we may just leave it for future work. I will raise my score.

---

### Author Rebuttal · Authors · 2023-08-10

We sincerely appreciate all reviewers’ time and efforts in reviewing our paper. In addition to the response to specific reviewers, here we would like to highlight our contributions and the new experiments that we add in the rebuttal.

&nbsp;

**[Our Contributions]**

We are glad to find out that the reviewers generally acknowledge our contributions:
 * The motivation is reasonable and clear. [KXNR, QPPK]
 * The idea of injecting the 3D world into large language models is novel,  valid and functioning. [G3Mn, 6ag4]
 * The technical contributions, including data generation, overall framework, and experimental analysis are solid and convincing. The 3D-Language dataset will be released. [G3Mn]
 * The experimental results cover a wide range of tasks, including 3D captioning, 3D question answering, task decomposition, 3D grounding, 3D-assisted dialog, and navigation. The results are substantial and solid. [QPPK]
 * The organization and writing of the paper are fluent.[KXNR, G3Mn, QPPK]

**[New Experiments]**

In this rebuttal, we have added more supporting experiments to address reviewers’ concerns.
* Results using existing pre-trained 3D encoders for 3D-LLMs [6ag4]
* Results on pre-trained 3D-LLMs without finetuning [KXNR, QPPK]
* Comparision with recent 3D captioning model [KXNR]
* Measurement of Hallucination [KXNR]
* New results on ScanRefer [KXNR]
* Comparison among 3D feature generation approaches [G3Mn]
* ScanQA baselines pretrained on 3D-language data [QPPK]

**[Qualitative examples]**

We attach two qualitative examples in the PDF:
* Results on ModelNet and KITTI [KXNR]
* Qualitative examples of comparison between OpenShape and 3D-LLM on captioning [KXNR]

&nbsp;

We hope our responses below convincingly address all reviewers’ concerns. We thank all reviewers’ time and efforts again!

---

### Author Response · Authors · 2023-08-12
**Thank you and we are looking forward to your post-rebuttal feedback!**

Dear AC and all reviewers:

Thanks again for all the insightful comments and advice, which helped us improve the paper's quality and clarity.

The discussion phase has been on for several days and we have not heard any post-rebuttal responses yet.

We would love to convince you of the merits of the paper. Please do not hesitate to let us know if there are any additional experiments or clarification that we can offer to make the paper better. We appreciate your comments and advice.

Best,

Author

---

> ### Comment · Area_Chair_rcuh · 2023-08-13
>
> Dear Reviewers,
>
> The authors have posted detailed individual responses to your review feedback, and the current reviewer-author discussion phase ends on Aug 21.
>
> Pls read the rebuttal(s) and respond within the **next 2-3 days**, so as to allow for further discussion (if needed).
>
> You should read the authors' rebuttal(s) in detail, and ideally:
> 1) Acknowledge the points that you find have been satisfactorily addressed;
> 2) Ask for further clarifications where needed;
> 3) Explain why you may still find certain issues to be insufficiently addressed
>
> Thanks,
> AC

---

### Decision · Program_Chairs · 2023-09-21

**Decision:**

Accept (spotlight)

**Comment:**

Reviewers were unanimously positive about this paper. There are a good number of definite strengths: clear, novel and very timely motivation; wide range of tasks covered; solid experimental results.

Nonetheless, there were also a number of issues/questions raised in the initial reviews, including baseline methods, number of tokens needed, potential data leakage, use of 2D multi-view images instead of "pure" 3D point cloud data, etc. However, the rebuttals included a good number of experimental results (e.g. zero-shot performance, baselines, etc.), addressing the issues raised, and reviewers were satisfied that their concerns were addressed. Perhap one issue that was not quite satisfactorily addressed in this work itself, is the issue of a true 3D point cloud encoder -- but I concur that this can be stated as a limitation and left for future work.

Overall, this is a solid and timely piece of work that deserves attention from the NeurIPS community.